# Sleep–wake patterns are altered with age, Prdm13 signaling in the DMH, and diet restriction in mice

Shogo Tsuji[1,*], Cynthia S Brace[2,*], Ruiqing Yao[1], Yoshitaka Tanie[1], Hirobumi Tada[1,3,4], Nicholas Rensing[5], Seiya Mizuno[6], Julio Almunia[7], Yingyi Kong[8], Kazuhiro Nakamura[9], Takahisa Furukawa[10], Noboru Ogiso[7], Shinya Toyokuni[8], Satoru Takahashi[6], Michael Wong[5], Shin-ichiro Imai[2,11], Akiko Satoh[1,12]

Old animals display significant alterations in sleep–wake patterns such as increases in sleep fragmentation and sleep propensity. Here, we demonstrated that PR-domain containing protein 13 (Prdm13)+ neurons in the dorsomedial hypothalamus (DMH) are activated during sleep deprivation (SD) in young mice but not in old mice. Chemogenetic inhibition of Prdm13+ neurons in the DMH in young mice promotes increase in sleep attempts during SD, suggesting its involvement in sleep control. Furthermore, DMH-specific *Prdm13*-knockout (DMH-*Prdm13*-KO) mice recapitulated age-associated sleep alterations such as sleep fragmentation and increased sleep attempts during SD. These phenotypes were further exacerbated during aging, with increased adiposity and decreased physical activity, resulting in shortened lifespan. Dietary restriction (DR), a well-known anti-aging intervention in diverse organisms, ameliorated age-associated sleep fragmentation and increased sleep attempts during SD, whereas these effects of DR were abrogated in DMH-*Prdm13*-KO mice. Moreover, overexpression of *Prdm13* in the DMH ameliorated increased sleep attempts during SD in old mice. Therefore, maintaining Prdm13 signaling in the DMH might play an important role to control sleep–wake patterns during aging.

## Introduction

The elderly commonly experiences changes in their sleep habits and sleep disruptions that causes problems, including waking often during the night, needing daytime naps, and having trouble falling asleep. The National Institute on Aging conducted a multicentered study called "established populations for epidemiologic studies of the elderly (EPESE)" with more than 9,000 participants aged 65 yr and older (Foley et al, 1995). Interestingly, people who reported excessive sleepiness during the afternoon or evening had a slight, but statistically significant increase in the odds for 3-yr mortality. Even in mice, recent studies have demonstrated that old C57BL/6J mice exhibit reduced amount of wakefulness and increased amount of non-rapid eye movement (NREM) sleep (Panagiotou et al, 2017; McKillop et al, 2018; Soltani et al, 2019). In both humans and mice, old individuals display sleep fragmentation, characterized by shorter episode durations of wakefulness, NREM, and REM sleep, compared with young individuals (Carskadon et al, 1982; Naidoo et al, 2008; Wimmer et al, 2013; Mander et al, 2017; Panagiotou et al, 2017; Soltani et al, 2019). Indeed, chronic sleep fragmentation is known to be associated with derailments in physiological functions, including low physical activity, increased adiposity, and metabolic dysfunction (Hakim et al, 2015; He et al, 2015). Thus, it is conceivable that the dysregulation of sleep–wake patterns has a mechanistic connection to age-associated physiological decline. However, such a mechanistic connection has remained elusive, and it is unclear whether any effective intervention could improve age-associated sleep dysfunction.

The hypothalamus plays a critical role in the regulation of sleep–wake patterns (Scammell et al, 2017) and aging and longevity in mammals (Satoh et al, 2013; Zhang et al, 2013; Zhang et al, 2017). In our previous study, we have demonstrated that the mammalian NAD[+]-dependent protein deacetylase Sirt1 in the dorsomedial and lateral hypothalami (DMH and LH, respectively) delays aging, with significant enhancement of physical activity, oxygen consumption, body temperature, and delta power, which is an indicator for depth of sleep and extends lifespan in mice (Satoh et al, 2013; Zhang et al, 2013; Zhang et al, 2017). Furthermore, knockdown of *Sirt1* in the DMH

[1]Department of Integrative Physiology, National Center for Geriatrics and Gerontology (NCGG), Obu, Japan   [2]Department of Developmental Biology, Washington University School of Medicine, St. Louis, MO, USA   [3]Department of Nutrition, Faculty of Wellness, Shigakkan University, Obu, Japan   [4]Department of Physiology, Yokohama City University Graduate School of Medicine, Yokohama, Japan   [5]Department of Neurology, Washington University School of Medicine, St. Louis, MO, USA   [6]Laboratory Animal Resource Center, University of Tsukuba, Tsukuba, Japan   [7]Laboratory of Experimental Animals, NCGG, Obu, Japan   [8]Department of Pathology and Biological Responses, Nagoya University Graduate School of Medicine, Nagoya, Japan   [9]Department of Integrative Physiology, Nagoya University Graduate School of Medicine, Nagoya, Japan   [10]Laboratories for Molecular and Developmental Biology, Institute for Protein Research, Osaka University, Osaka, Japan   [11]Department of Gerontology, Laboratory of Molecular Life Science, Institute of Biomedical Research and Innovation, Kobe, Japan   [12]Department of Integrative Physiology, Institute of Development, Aging, and Cancer, Tohoku University, Sendai, Japan

Correspondence: asatoh@ncgg.go.jp
*Shogo Tsuji and Cynthia S Brace contributed equally to this work

and LH causes low delta power, and another mouse model with high hypothalamic Sirt1 activity displays reduced sleep fragmentation with advanced age and lifespan extension (Yoshida et al, 2019). These findings suggested a possibility that a specific sub-population of Sirt1+ neurons in the DMH and/or LH controls sleep–wake patterns during the process of aging. We previously conducted a comprehensive transcriptome analysis to identify DMH-enriched genes and identified *PR-domain containing protein 13 (Prdm13)*. Importantly, *Prdm13* is regulated by Sirt1 signaling, and DMH-specific *Prdm13* knockdown mice show low delta power (Satoh et al, 2015). Therefore, we hypothesized that Prdm13 signaling in the DMH, where Sirt1 signaling is involved in aging and longevity control, is causally involved in age-associated sleep alterations.

In the present study, to address this hypothesis, we generated DMH-specific *Prdm13*-knockout (DMH-*Prdm13*-KO) mice and found that DMH-*Prdm13*-KO mice display sleep fragmentation and excessive sleepiness during sleep deprivation (SD), which are common phenomena in aged C57BL/6J mice. Aging DMH-*Prdm13*-KO mice displayed further exaggerated sleep alterations, increased adiposity, decreased physical activity, and shortened lifespan. Chemogenetic inhibition of Prdm13+ neurons in the DMH in young mice promotes increase in sleep attempts during SD, suggesting its involvement in sleep control. We also found that dietary restriction (DR), a well-known anti-aging intervention in diverse organisms (Fontana et al, 2010), ameliorates age-associated derailment of sleep–wake patterns. These effects of DR were abrogated in DMH-*Prdm13*-KO mice. Moreover, overexpression of *Prdm13* in the DMH ameliorated excessive sleepiness during SD in old mice. Thus, our results suggest that Prdm13 is involved in the regulation of sleep–wake patterns by DR.

## Results

### Old mice showed increases in sleep fragmentation and sleep propensity compared with young mice

Sleep fragmentation is one of the most common clinical characteristics in old individuals in both humans and mice (Carskadon et al, 1982; Naidoo et al, 2008; Wimmer et al, 2013; Mander et al, 2017; Panagiotou et al, 2017; Soltani et al, 2019). To confirm age-associated sleep fragmentation, we conducted electroencephalogram (EEG) and electromyogram (EMG) recordings in young and old mice at 4 and 20 mo of age, respectively. Old mice disp mice were significantly higher than young mice (Fig 1A), whereas the duration of NREM sleep episodes in old mice was shorter than young mice (Fig 1B). The duration of wakefulness and REM sleep episodes in old mice was also shorter during the light period (Fig 1B). In addition, in a 24-h period, old mice spent less time awake and more time in NREM sleep (Figs 1C and S1A). As most mouse studies reported (Wimmer et al, 2013; Panagiotou et al, 2017; McKillop et al, 2018; Soltani et al, 2019), the total amount of wakefulness in old mice was significantly lower than young controls during the dark period, whereas the total amount of NREM sleep was higher (Fig 1C). Together, our

data confirm that old mice display greater sleep fragmentation and spend more time asleep compared with young mice.

To examine the profile of EEG spectra during each state in young and old mice, we used fast Fourier transform (FFT) of EEG recordings. During wakefulness, the power of EEG in the frequency range between 4.3 and 12 Hz in old mice was significantly lower than young mice (repeated measures ANOVA: factor age $F_{(1,9)}$ = 6.455, $P$ = 0.0317) (Fig 1D). Because the activity of the theta frequency range during wakefulness is correlated with arousal (Wimmer et al, 2013; Panagiotou et al, 2017), old mice might have reduced arousal and less exploratory behavior compared with young mice. This result is consistent with the finding that old mice display an increased sleep propensity (Fig 1C). The spectral power of the delta frequency range during NREM sleep is known as slow wave activity (SWA) and a good indicator of sleep intensity (Wimmer et al, 2013). It has been reported that the absolute value of EEG SWA is significantly increased (Panagiotou et al, 2017; McKillop et al, 2018) or tended to be increased (Wimmer et al, 2013) in old mice. In our study, the power of the NREM EEG in the frequency range between 2.3 and 6.3 Hz in old mice was higher than young mice, but this trend did not reach statistical significance (Fig 1D). The absolute value of SWA in old mice during a 24-h period also tended to be increased compared with young mice (Fig 1E). It has been suggested that absolute levels of SWA correlate with sleep pressure (Panagiotou et al, 2017), thus old mice might be exposed daily to a high sleep pressure compared with young mice. Although some studies showed significantly lower theta peak of REM EEG spectra in old mice, no notable differences were found in the REM EEG spectra in the frequency range between 4 and 9 Hz in our study (Fig 1D).

### Old mice display increased sleep attempts during SD and homeostatic sleep response to SD

We next evaluated whether aging affects homeostatic sleep response by examining responses to SD in young and old mice. Sleep was disrupted by gentle handling for 6 h, and then the mice were allowed to recover sleep loss (Fig 1F). The number of sleep attempts gradually increased during SD in both young and old mice, and we noticed that old mice showed excessive sleepiness as their sleep attempts during SD were much greater than young mice (repeated measures ANOVA: factor age $F_{(1,9)}$ = 5.989, $P$ = 0.0369) (Fig 1G). Thus, these results suggest that old mice might be more susceptible to accumulate sleep pressure from sleep loss than young mice. On the other hand, both young and old mice displayed a significant increase in SWA after SD (Figs 1H and S1B), indicating that the homeostatic response to SD is intact in old mice, which is consistent with other recent literature (Wimmer et al, 2013; Panagiotou et al, 2017; McKillop et al, 2018). Surprisingly, the level of initial increase of SWA after SD in old mice was significantly higher than young mice (repeated measures ANOVA: factor age × time $F_{(5,45)}$ = 7.162, $P$ < 0.0001), further supporting the notion that old mice might accumulate more sleep pressure during SD.

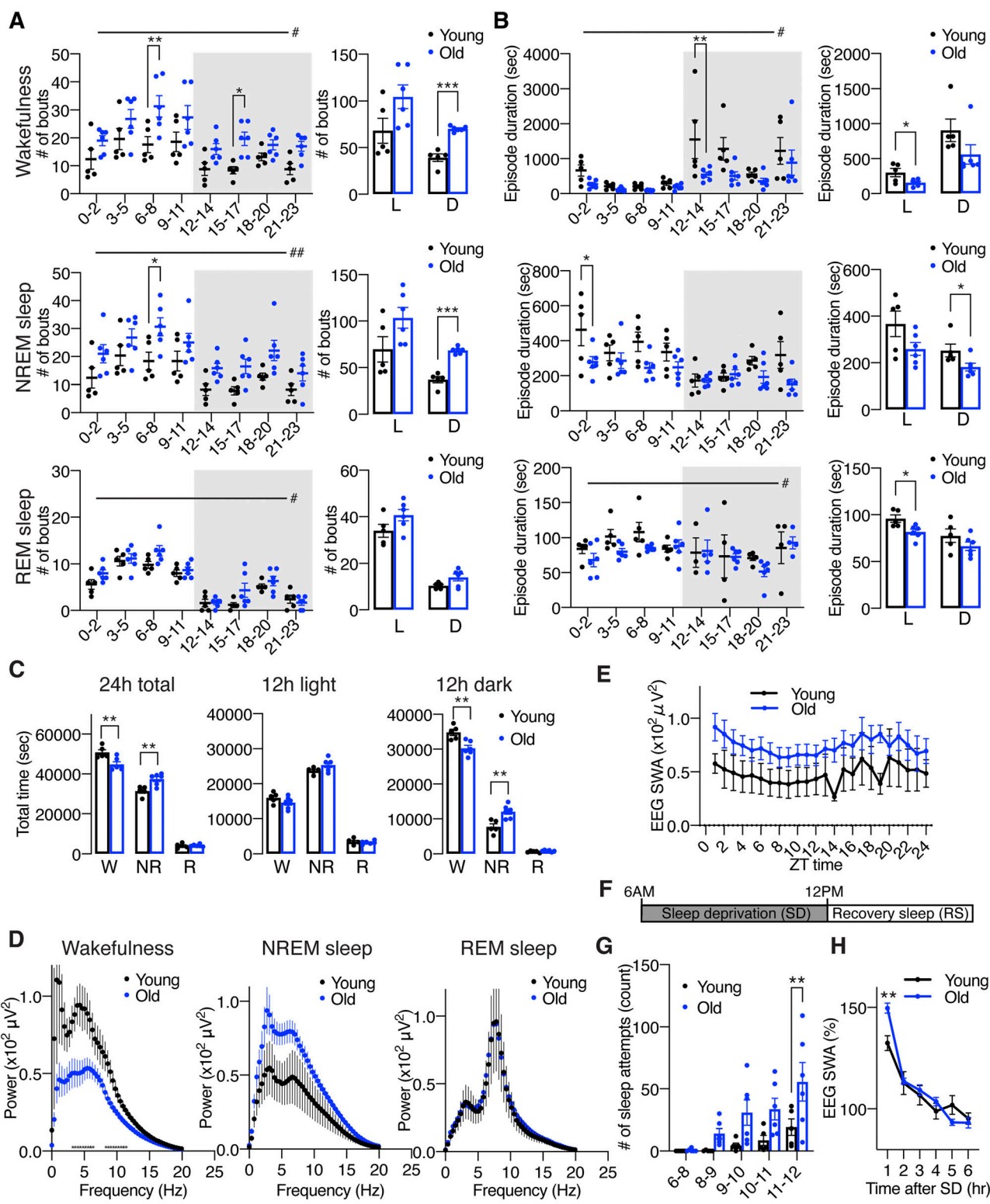

**Figure 1. Old C57BL/6J mice display increases in sleep fragmentation, sleep propensity, and excessive sleepiness during SD.**
**(A, B)** Numbers of episodes (A) and duration (B) of wakefulness (top), NREM sleep (middle), and REM sleep (bottom) every 3 h through a day (left) and during the light (L) and dark (D) periods (right) in young and old mice (n = 5–6). Shading indicates dark period. Values are shown as means ± S.E., #P < 0.05 and ##P < 0.01 by repeated measures ANOVA, *P < 0.05, **P < 0.01, and ***P < 0.001 by repeated measures ANOVA with Bonferroni's post hoc test (left) or unpaired t test (right). **(C)** Total amount of wakefulness, NREM sleep, and REM sleep during a 24-h period (24 h total), 12-h light period (12 h light) or 12-h dark period (12 h dark) (n = 5–6). Values are shown as means ± S.E., listed P-values, **P < 0.01 by unpaired t test. **(D)** EEG spectra of wakefulness (left), NREM sleep (middle), and REM sleep (right) during the light period (n = 5–6). Values are shown as means ± S.E. **(E)** SWA in the range of frequencies between 0.5 and 4 Hz during NREM sleep for a 24-h period (n = 5–6). Values are shown as means ± S.E. *P < 0.01 by

## The number of cFos+ cells in brain regions involved in the regulation of arousal and sleepiness increases significantly during SD

We examined which brain regions mainly responded to SD by staining the cFos protein, an immediate early gene product and a marker of neuronal activation (Gao & Ji, 2009), in brain sections collected during SD, recovery sleep (RS), and control sleep (SD-Cont and RS-Cont) (Fig 1F). During SD, we found that the number of cFos+ cells was elevated in the brain regions known to regulate arousal and sleepiness, including the hypothalamus and brainstem. In the hypothalamus, the DMH (summed up from bregma −1.67 to −1.91 mm) showed a greater number of cFos+ cells during SD compared with SD-Cont (Figs 2A–C and S2A). The median preoptic nucleus (MnPO) also showed increases in cFos+ cells during SD (Fig S2B) but not statistically significant. The number of cFos+ cells was significantly suppressed during RS in the DMH and MnPO compared with SD (Figs 2A and S2B). The LH, ventrolateral preoptic nucleus (VLPO), and tuberomammillary nucleus exhibited no differences between SD-Cont and SD, although the number of cFos+ cells was significantly suppressed during RS in the LH and tuberomammillary nucleus compared with SD (Fig S2B). Therefore, neurons in the DMH and MnPO are activated specifically in response to sleep loss during SD, whereas cFos+ cells in response to SD have been reported in the MnPO (Brown et al, 2012; Alam et al, 2014), and those in the DMH have been poorly characterized. Although DMH neurons are linked to sleep (Zhong et al, 2022), aging and longevity control (Satoh et al, 2013; Zhang et al, 2013; Zhang et al, 2017), and also activated by psychological stress (Kataoka et al, 2014), the involvement of DMH neurons in sleep control has not been fully elucidated. Thus, we decided to focus on these cFos+ cells in the DMH in response to SD.

### Prdm13+ neurons in the DMH are activated in response to SD

Given that *Prdm13* is one of the DMH-enriched genes and involved in sleep regulation (Satoh et al, 2015), we suspected that the cFos+ DMH cells responding to SD would include Prdm13+ neurons.

To visualize Prdm13+ cells, *Prdm13*-CreERT2 mice were produced by targeted insertion of the coding sequence of tamoxifen-inducible Cre recombinase and 2A peptide into the native 3′ end of the *Prdm13* gene, generating the Prdm13-2A-CreERT2 protein. By crossing *Prdm13*-CreERT2 mice to Cre-dependent ZsGreen reporter mice, *Prdm13+* cells were visualized in the DMH and other brain regions such as the tuberal nucleus (TN) and amygdala (Amg) (Fig 2D). No ZsGreen expression was observed without tamoxifen (data not shown). In situ hybridization confirmed ZsGreen+ cells were co-localized with endogenous *Prdm13* mRNA (Fig S2C). We also investigated electrophysiological characteristics of Prdm13+ DMH cells by whole-cell patch-clamp technique. Using *Prdm13*-CreERT2-ZsGreen mice at 5–6 mo of age, ZsGreen+ (Prdm13+) cells in the compact area of the DMH were selected (Fig S2D). We recorded

synaptic activity (Fig S2E) and membrane capacitance (Cm), which is correlated with the morphology of neurons (Liu et al, 1996) (Fig S2F), and confirmed that Prdm13+ DMH cells are electrically active cells, such as a neuron (Fig S2G). Importantly, RNAscope analysis, a highly sensitive in situ hybridization method, revealed that the percentage of *cFos+* cells among *Prdm13+* DMH neurons was significantly higher during SD than SD-Cont (Fig 2E–G). Thus, Prdm13+ neuronal population in the DMH responds to sleep loss during SD in young mice. On the other hand, the percentage of *cFos+* cells in *Prdm13+* DMH neurons during SD did not differ from its percentage during SD-Cont in old mice (Figs 2H and S2H). These results indicate that normal neuronal activation of *Prdm13+* DMH neurons to SD is impaired with aging.

### Chemogenetic inhibition of Prdm13+ DMH neurons induces excessive sleepiness during SD

Next, using *Prdm13*-CreERT2 mice and the inhibitory designer receptor exclusively activated by designer drug hM4Di, we aimed to acutely inhibit the activity of Prdm13+ DMH neurons during SD (Fig 3A). AAV8-hSyn-DIO-hM4Di-mCherry was bilaterally injected into the DMH of heterozygous *Prdm13*-CreERT2 (*Prdm13*^CreERT2/+) or control (*Prdm13*^+/+) mice, and Cre-dependent expression of DIO-hM4Di-mCherry within the DMH was confirmed after tamoxifen treatment (Fig 3B). When hM4Di-expressing *Prdm13*^CreERT2/+ mice were subjected to SD, the mice injected with clozapine-N-oxide (CNO) showed significantly greater number of sleep attempts within 60 min (from 9 AM to 10 AM) than those given vehicle injection (repeated measures ANOVA: $F_{(1,20)} = 7.744$, $P = 0.0115$) (Fig 3C left). In addition, the total number of sleep attempts in hM4Di-expressing *Prdm13*^CreERT2/+ mice were significantly increased for 2 h after CNO injection (Fig 3C right). On the other hand, the number of sleep attempts during SD in *Prdm13*^+/+ mice was indistinguishable between CNO and vehicle treatment (repeated measures ANOVA: $F_{(1,13)} = 2.261$, $P = 0.1565$) (Fig 3D). These results demonstrate that the suppression of Prdm13+ DMH neurons induces excessive sleepiness during SD. The level of SWA after SD was similar between treatment groups (Figs 3E and S3), and the number of sleep attempts in the last 1 h of SD (from 10 AM to 11 AM) in CNO treatment did not significantly differ from vehicle treatment (Fig 3C). Thus, the level of sleep pressure immediately after SD was comparable between groups.

### Mice with deficiency of *Prdm13* in the DMH display sleep fragmentation and excessive sleepiness during SD

To elucidate the role of Prdm13 signaling in age-associated sleep alterations, we generated DMH-*Prdm13*-KO mice. Our previous study demonstrated that *Prdm13* expression is partially regulated by Nkx2-1, which is highly expressed in the DMH (Satoh et al, 2015). We confirmed that most of Prdm13 is co-expressed with Nkx2-1 in the DMH but not in the TN and Amg (Fig S4A). The percentage of Prdm13+Nkx2-1+ cells within Prdm13+ cells was 60% ± 7.6%, 71% ± 6.1%,

unpaired *t* test. (F) Schematic of SD. Sleep was deprived for 6 h, between 6 AM and 12 PM, followed by a period of RS. (G) Number of sleep attempts during SD from 6 AM to 8 AM (6–8), 8 AM to 9 AM (8–9), 9 AM to 10 AM (9–10), 10 AM to 11 AM (10–11), and 11 AM to 12 PM (11–12) in young and old mice (n = 5–6). Values are shown as means ± S.E., **$P$ < 0.01 and by repeated measures ANOVA with Bonferroni's post hoc test. (H) SWA after SD in young and old mice (n = 5–6). Each value is relative to the average of the 24-h baseline day. Values are shown as means ± S.E., **$P$ < 0.01 by repeated measures ANOVA with Bonferroni's post hoc test.

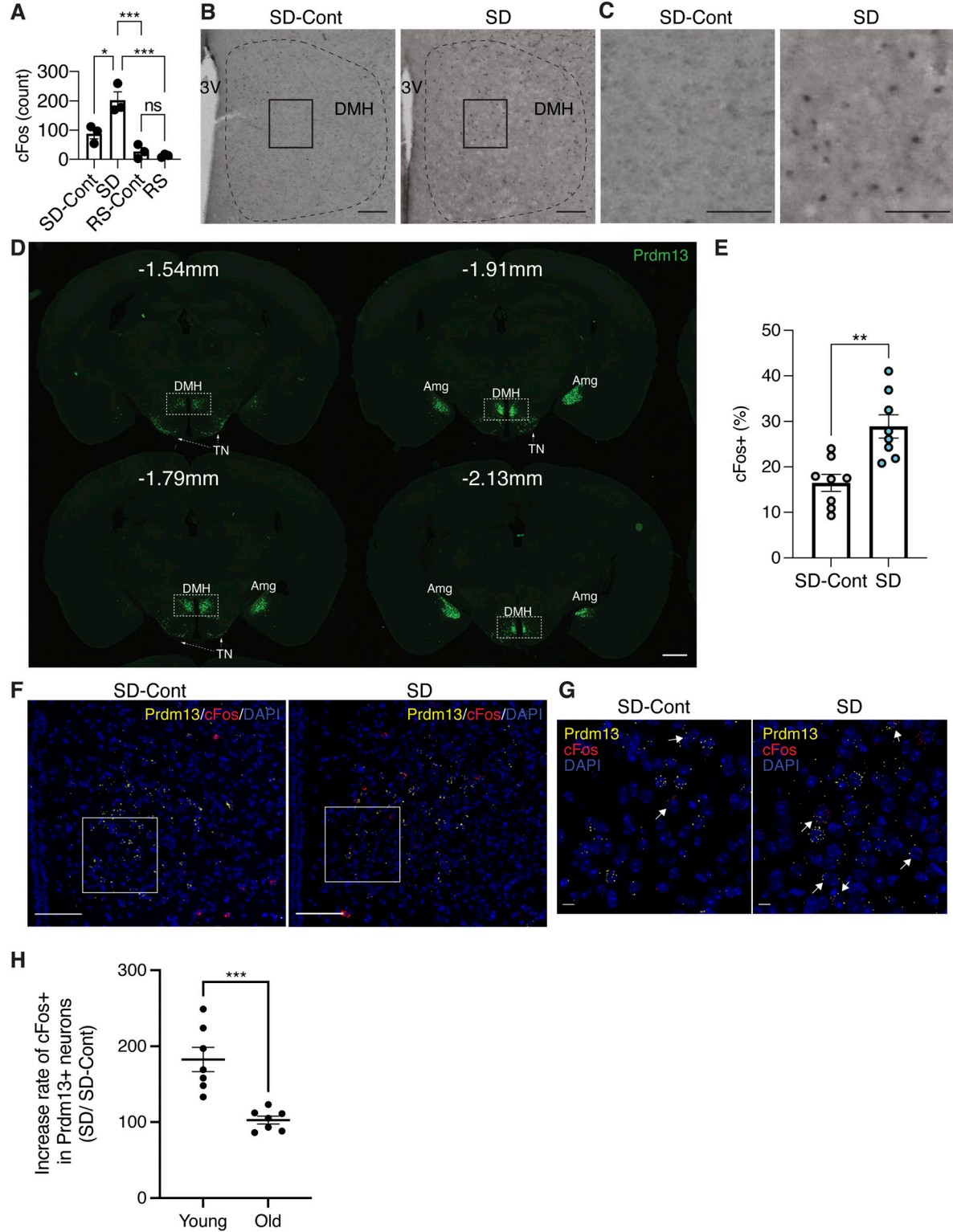

**Figure 2. Prdm13+ neurons in the DMH are activated during SD.**
**(A)** Total numbers of cFos+ cells in the DMH during SD, RS, and sleeping control (SD-Cont, RS-Cont) detected by cFos immunohistochemistry. The total number of cFos+ cells in the DMH was counted at bregma −1.67 mm, −1.79 mm, and −1.91 mm and summed up (total three sections) each mouse (n = 3). The third ventricle (3V) is shown. Values are shown as means ± S.E., *$P < 0.05$, ***$P < 0.001$, and non-significant (ns) by one-way ANOVA with Bonferroni's post hoc test. **(B, C)** Images of DMH sections at bregma −1.67 mm from mice under SD-Cont (left) and SD (right) with cFos. Boxed areas were shown at high magnification in (C). Scale bar indicates 100 and 25 $\mu$m (B and C, respectively). **(D)** Images of the ZsGreen signal (Prdm13, green) including the DMH, amygdala (Amg), and tuberal nucleus (TN) at bregma −1.54, −1.79, −1.91, and −2.13 mm of *Prdm13*-ZsGreen mice. Scale bar indicates 100 $\mu$m. **(E)** Ratios of *cFos+* cells within *Prdm13+* cells in young mice during SD and SD-Cont detected by RNAscope in situ

and 81% ± 3.9% at bregma −1.43 mm to −1.67 mm, −1.79 mm, and −1.91 mm, respectively (Fig S4B). Thus, we crossed *Prdm13*-floxed mice with *Nkx2-1*-CreERT2 mice to generate DMH-*Prdm13*-KO mice (Fig 4A). The knockout efficiency of *Prdm13* in the DMH was about 70% after tamoxifen induction (Fig 4B). Significant reduction of *Prdm13* expression was not observed in the TN and Amg of DMH-*Prdm13*-KO mice (Fig 4B), and this event was specific to the hypothalamus because the expression of *Prdm13* remained intact in the retina where *Prdm13* is highly expressed (Fig S4C).

We then analyzed sleep–wake patterns in DMH-*Prdm13*-KO and control mice at 4 mo of age. During the light period between ZT0 to ZT5, DMH-*Prdm13*-KO mice showed a tendency of increase in the numbers of wakefulness and NREM sleep episodes compared with control mice (wakefulness; repeated measure ANOVA: factor genotype $F_{(1,10)}$ = 4.796, $P$ = 0.053, NREM sleep; repeated measure ANOVA: factor genotype $F_{(1,10)}$ = 3.539, $P$ = 0.089) (Fig 4C). The duration of wakefulness episodes in DMH-*Prdm13*-KO mice was significantly shorter than control mice during the light period between ZT0 to ZT2. The duration of NREM sleep episodes in DMH-*Prdm13*-KO mice was significantly shorter than control mice during the dark period (Fig 4D). These results indicate that DMH-*Prdm13*-KO mice showed sleep fragmentation compared with control mice. We also assessed their responses to SD. The number of sleep attempts during SD in DMH-*Prdm13*-KO mice was significantly higher than those in control mice (repeated measure ANOVA: factor genotype $F_{(1,25)}$ = 9.131, $P$ = 0.0057) (Fig 4E), recapitulating the phenotype of old WT mice (Fig 1G). The level of initial increase of SWA after SD in DMH-*Prdm13*-KO mice was similar to control mice (Figs 4F and S4D), suggesting that the level of sleep pressure is comparable to each other. During wakefulness, the power of the EEG spectra at the frequency range between 4 and 12 Hz, in particular 4–9 Hz, in DMH-*Prdm13*-KO mice tended to be lower than control mice (repeated measures ANOVA: factor genotype $F_{(1,9)}$ = 0.7446, $P$ = 0.4106) (Fig 4G), but there was no statistical significance. This trend was observed in old WT mice (Fig 1D). The absolute value of SWA in DMH-*Prdm13*-KO mice during a 24-h period tended to be higher than young mice (Fig S4E) but was not statistically significant. There were no abnormalities in the amounts of sleep and wakefulness, circadian period length, and wheel-running activity in DMH-*Prdm13*-KO mice (Figs 4H and S4F–H). Together, sleep fragmentation during NREM sleep and excessive sleepiness during SD are commonly observed in old mice and DMH-*Prdm13*-KO mice, but the effects of aging and Prdm13-KO on sleep propensity were distinct from each other. Similarly, a mouse model with high hypothalamic Sirt1 activity displays reduced number of transitions between wakefulness and NREM sleep (Yoshida et al, 2019), revealing that hypothalamic Sirt1, and Prdm13, is involved in the regulation of sleep fragmentation. Sleep propensity was not altered in brain-specific *Sirt1*-overexpressing transgenic mice (Satoh et al, 2013). Given that the level of hypothalamic *Prdm13* and its function decline with age (Satoh et al, 2015), age-associated sleep fragmentation could be promoted through the reduction of Prdm13/Sirt1 signaling in the DMH, but sleep propensity might be increased by other mechanisms.

## Old DMH-*Prdm13*-KO mice display increased sleep fragmentation, adiposity, low physical activity, and short lifespan compared with old control mice

To address the possibility that DMH-*Prdm13*-KO mice might accelerate physiological changes with advanced age, we conducted additional assessments using DMH-*Prdm13*-KO and control mice at 20 mo of age. The level of sleep fragmentation in old DMH-*Prdm13*-KO mice was significantly higher than old control mice (Fig 5A and B). In old DMH-*Prdm13*-KO mice, the number of wakefulness and NREM sleep episodes were significantly higher during the light period (Fig 5A), and episode duration of wakefulness was significantly shorter (Fig 5B) compared with old control mice. The duration of NREM sleep episodes in old DMH-*Prdm13*-KO mice was significantly shorter during the dark phase compared with old control mice (Fig 5B). Therefore, long-term deficiency of Prdm13 signaling in the DMH worsens sleep fragmentation, particularly during the light period. The power of the EEG spectra at the frequency between 4 and 12 Hz, in particular 6.6–12 Hz, during wakefulness in old DMH-*Prdm13*-KO mice was significantly lower than old control mice (repeated measures ANOVA: factor genotype $F_{(1,8)}$ = 5.337, $P$ = 0.0497) (Fig 5C), suggesting that old DMH-*Prdm13*-KO mice display increased sleep propensity compared with old control mice. No differences were found in the NREM and REM EEG spectra (Fig 5C). The absolute value of SWA in old DMH-*Prdm13*-KO mice during a 24-h period was tended to be higher than young mice (Fig S5A), but there was no statistical significance. Old DMH-*Prdm13*-KO mice displayed excessive sleepiness during SD compared with old controls (repeated measures ANOVA: factor genotype $F_{(1,9)}$ = 5.341, $P$ = 0.0462) (Fig 5D). The level of initial increase of SWA after SD in old DMH-*Prdm13*-KO mice was significantly higher than old control mice (repeated measures ANOVA: factor time × genotype $F_{(5,45)}$ = 5.024, $P$ = 0.0010) (Figs 5E and S5B). Therefore, old DMH-*Prdm13*-KO mice presumably accumulated more sleep pressure during SD compared with old control mice. The circadian period length and the amount of sleep and wakefulness were indistinguishable between old DMH-*Prdm13*-KO and control mice (Fig S5C–E), suggesting that circadian function, one of the major factors governing sleep–wake patterns (Scammell et al, 2017), was still intact in old DMH-*Prdm13*-KO mice. Although there was no change in body weight between DMH-*Prdm13*-KO and control mice at young age, DMH-*Prdm13*-KO mice gained more body weight than control mice at 20 mo of age (Fig 5F). The weight of perigonadal white adipose tissue in old DMH-*Prdm13*-KO mice tended to be higher than control mice ($P$ = 0.079 by unpaired $t$ test) (Fig S5F), and the size of adipocyte was significantly larger than control mice (Fig S5G and H). Moreover, the level of physical activity in old DMH-*Prdm13*-KO mice was significantly lower than old control mice (repeated measures ANOVA: factor genotype $F_{(1,10)}$ = 8.842,

hybridization (n = 8). Values are shown as means ± S.E., *$P$ < 0.05 and **$P$ < 0.01 by two-way ANOVA with Bonferroni's post hoc test. **(F, G)** Representative images of DMH sections from young mice under SD-Cont (left) and SD (right) with *Prdm13* (yellow) and *cFos* (red) visualized by RNAscope. Boxed areas were shown at high magnification in (G). Cells were counterstained with DAPI (blue). Scale bars indicate 100 and 10 μm (F and G, respectively). **(H)** Ratios of cFos+ cells within Prdm13+ cells under SD relative to SD-Cont in young and old mice (n = 7–8). Values are shown as means ± S.E., ***$P$ < 0.001 by two-way ANOVA with Bonferroni's post hoc test.

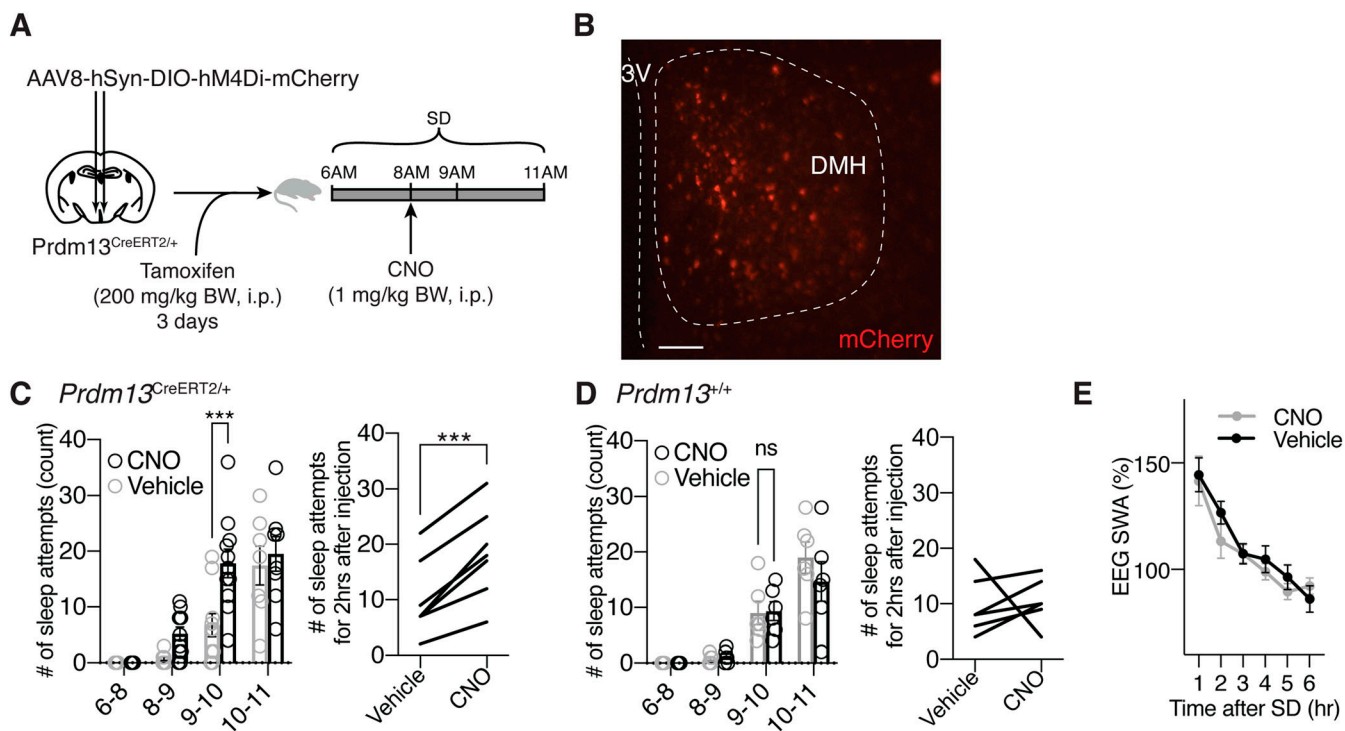

**Figure 3. Chemogenetic inhibition of Prdm13+ DMH neurons promotes excessive sleepiness during SD.**
**(A)** Experimental procedure for chemogenetic inhibition of Prdm13+ DMH neurons. AAV-hSyn-DIO-hM4Di-mCherry virus was injected into the DMH of *Prdm13*^CreERT2/+ mice, and tamoxifen was intraperitoneally injected for three consecutive days (200 mg/kg body weight, i.p.). During SD between 6 AM and 11 AM, clozapine-N-oxide (CNO) was intraperitoneally injected at 8 AM (1 mg/kg body weight). **(B)** A representative image of the injection site (mCherry, red) in the DMH, near the 3 V. Scale bar indicates 100 μm. **(C, D)** Numbers of sleep attempts during SD from 6 AM to 8 AM (6–8), 8 AM to 9 AM (8–9), 9 AM to 10 AM (9–10), and 10 AM to 11 AM (10–11) in mice with CNO or vehicle injection in *Prdm13*^CreERT2/+(C left) and *Prdm13*^+/+(D left) mice (n = 6–12). Values are shown as means ± S.E., ***P < 0.001 and non-significant (ns) by repeated measures ANOVA with Bonferroni's post hoc test. The number of sleep attempts within 2 h after CNO or vehicle injection in individual *Prdm13*^CreERT2/+ (C right) and *Prdm13*^+/+ (D right) mice. ***P < 0.001 by paired *t* test was obtained for the mean number of sleep attempts within 2 h between CNO and vehicle injection in *Prdm13*^CreERT2/+ mice (C right) (n = 7) but not in *Prdm13*^+/+ mice (D right) (n = 6). **(E)** SWA after SD of *Prdm13*^CreERT2/+ mice (n = 5). Normalized power is relative to the average of the 24-h baseline day. Values are shown as means ± S.E.

*P* = 0.014) (Fig 5G), although there was no change in food intake (Fig S5I). Taken together, DMH-*Prdm13*-KO mice exhibited the exacerbation of physiological decline with advanced age. Consistent with these observations, DMH-*Prdm13*-KO mice shortened their lifespan (*P* = 0.0178 by log-rank test) (Fig 5H). Because malignant neoplasm is the main cause of death in C57BL/6J mice (Blackwell et al, 1995), we next tested whether Prdm13 deficiency in the DMH affects the incidence of malignant neoplasm. Most of the DMH-*Prdm13*-KO and control mice died by malignant neoplasm (83% and 86%, respectively), revealing that deletion of *Prdm13* in the DMH does not directly affect age-associated malignancy. Sleep alterations cause age-associated physiological dysfunctions (Wang et al, 2014; Hakim et al, 2015; He et al, 2015). Thus, alterations of sleep–wake patterns due to the deficiency of *Prdm13* may accelerate the decline in certain physiological function and reduce life expectancy without affecting age-associated malignancy.

## Short-term DR ameliorates age-associated sleep alterations in the presence of Prdm13 signaling

DR has been well known to ameliorate a wide variety of age-associated pathophysiological dysfunctions, delaying aging, and extending lifespan. Thus, we speculated that DR could also ameliorate age-associated sleep alterations observed in old mice. After gradually decreasing the amount of food to 60% of daily food intake, mice at 20 mo of age were kept under DR for 14–28 d (Fig 6A). Control mice were fed ad libitum (AL). To minimize the disruption of their daily activity pattern, we fed mice at 5–6 PM right before the dark period for both DR and AL mice. Every day feeding did not cause disruption of the daily activity rhythm in AL and DR mice, except for some food-anticipatory activity (Stephan, 2002; Pendergast & Yamazaki, 2018) observed right before lights-off in DR mice (Fig S6A), resulting in increased total amount of wakefulness in DR mice during the light period (Fig S6B). However, the effect of DR on age-associated sleep propensity was not observed. Body weights in old DR mice were significantly lower than old AL mice at 14, 21, and 28 d after dietary intervention (Fig S6C). Remarkably, the number of wakefulness and NREM sleep episodes in old DR mice were significantly lower than old AL mice during the light and dark periods, and the number of REM sleep episodes in old DR mice were significantly lower (Fig 6B), whereas the durations of wakefulness, NREM, and REM sleep episodes were longer than old AL mice (Fig 6C). Intriguingly, the power of EEG spectra across the overall frequency range during NREM and REM sleep in DR mice were significantly lower than AL mice (NREM sleep: repeated measures ANOVA: factor diet $F_{(1,8)}$ = 10.25, *P* = 0.0126, REM: repeated

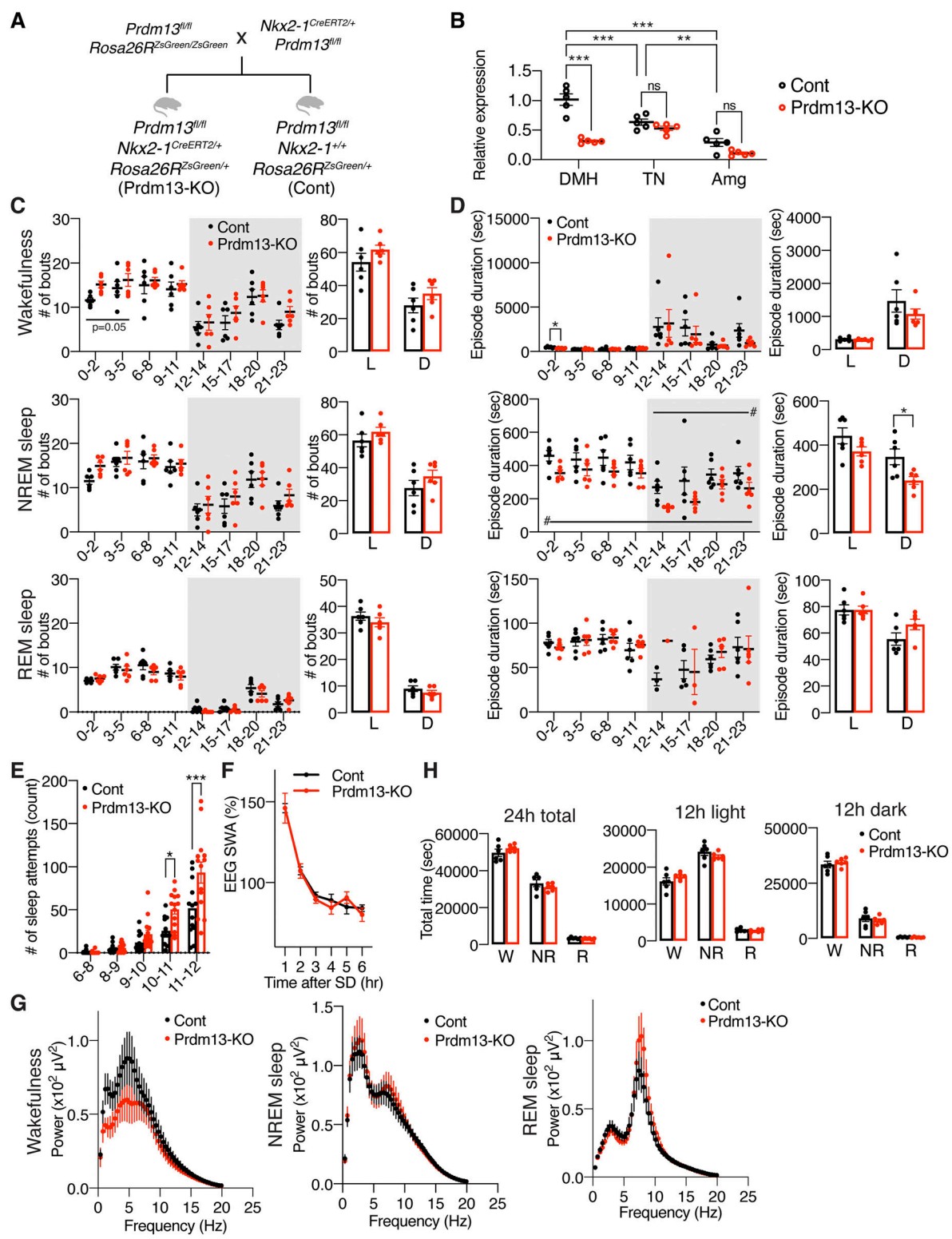

**Figure 4. DMH-specific *Prdm13*-knockout mice display sleep alterations observed in aged C57BL/6J mice.**
**(A)** Breeding strategy to generate DMH-specific *Prdm13*-knockout (Prdm13-KO) mice. After crossing *Prdm13 $^{fl/f}$; Rosa26R$^{ZsGreen/ZsGreen}$* mice and *Nkx2-1$^{CreERT2/+}$; Prdm13 $^{fl/fl}$* mice, *Prdm13 $^{fl/fl}$; Nkx2-1$^{CreERT2/+}$; Rosa26R$^{ZsGreen/+}$* mice were used as Prdm13-KO mice and *Prdm13 $^{fl/fl}$; Nkx2-1$^{+/+}$; Rosa26R$^{ZsGreen/+}$* mice were used as control (Cont) mice.
**(B)** Expression of *Prdm13* in the DMH, tuberal nucleus (TN), and amygdala (Amg) of Prdm13-KO and Cont mice (n = 5). Values are shown as means ± S.E., **$P < 0.01$, ***$P < 0.001$, and non-significant (ns) by two-way ANOVA with Bonferroni's post hoc test. **(C, D)** Number of episodes (C) or duration (D) of wakefulness (top), NREM sleep (middle), and REM sleep (bottom) every 3 h through a day (left) and during the light (L) and dark (D) periods (right) in Prdm13-KO and Cont mice. Shading indicates dark period (n = 6). Values are shown as means ± S.E., #$P < 0.05$ by repeated measures ANOVA, listed *P*-values and *$P < 0.05$ by repeated measures ANOVA with Bonferroni's post

measures ANOVA: factor diet $F_{(1,8)}$ = 5.405, $P$ = 0.0486) (Fig 6D). It has been reported that starvation promotes significant reduction of EEG spectra due to hypothermia (Huang et al, 2021). In fact, DR mice displayed significantly lower body temperature than AL mice over the course of experimental period (data not shown). Thus, decreases in EEG spectra in DR mice during NREM and REM sleep might be due to a low body temperature, not necessarily reflecting sleep pressure. The level of absolute SWA was significantly lower for a 24-h period in DR mice but increased during a mealtime around ZT12 (Fig S6D), when body temperature is elevated, further supporting the idea that body temperature is associated with the regulation of EEG spectral power. In addition, DR significantly suppressed the number of sleep attempts during SD (repeated measures ANOVA: factor diet $F_{(1,9)}$ = 5.131, $P$ = 0.0498) (Fig 6E). The level of SWA seen immediately after SD in DR mice was significantly lower than AL mice (repeated measures ANOVA: factor time × age $F_{(5,39)}$ = 11.70, $P$ < 0.0001) (Figs 6F and S6E), suggesting that DR mice have less sleep pressure after SD compared with AL mice. Notably, SWA was increased in DR mice compared with AL mice at 6 h after SD (ZT11) when SWA increased under the basal condition (Fig S6D), suggesting that daily rhythm of SWA is strongly persistent after DR. Taken together, DR effectively ameliorated age-associated sleep fragmentation and excessive sleepiness during SD.

Importantly, in DMH-*Prdm13*-KO mice that recapitulate the phenotypes of old WT mice, the effects of DR on sleep fragmentation and sleep attempts during SD were abrogated (Figs 6G and S6F and G). As expected, body weight was significantly lower in DMH-*Prdm13*-KO mice under DR compared with the same KO mice under AL-feeding, confirming that DR is properly conducted (Fig S6H). The number and durations of wakefulness and NREM sleep episodes in DMH-*Prdm13*-KO mice were indistinguishable between AL and DR (Fig S6F and G). On the other hand, the number of REM sleep episodes in DMH-*Prdm13*-KO under DR was significantly lower during the light period, but higher during the dark period than DMH-*Prdm13*-KO under AL (Fig S6F). In addition, the number of sleep attempts during SD was also indistinguishable between DR and AL in DMH-*Prdm13*-KO mice (repeated measures ANOVA: factor diet $F_{(1,14)}$ = 2.918, $P$ = 0.1097) (Fig 6G). Except for food-anticipatory activity observed right before lights-off in DMH-*Prdm13*-KO-DR mice, the amount of sleep and wakefulness were indistinguishable between DMH-*Prdm13*-KO mice under DR and DMH-*Prdm13*-KO mice under AL (Fig S6I). Together, these results strongly suggest that Prdm13 is necessary to promote DR effects on sleep.

### Overexpression of *Prdm13* in the DMH partially affects age-associated sleep alterations

Given that the level of hypothalamic *Prdm13* and its function decline with age (Satoh et al, 2015), we next questioned whether overexpression of *Prdm13* in the DMH affects age-associated sleep alterations. We bilaterally injected lentivirus carrying a full-length *Prdm13* cDNA into the DMH of mice at 22 mo of age. The level of *Prdm13* mRNA in DMH-specific *Prdm13*-overexpressing (*Prdm13*-OE) mice was 5–17-fold higher compared with control mice (Fig 6H). The number of wakefulness and NREM sleep episodes in old *Prdm13*-OE mice was significantly lower, whereas duration of wakefulness in old *Prdm13*-OE mice tended to be longer than old control mice during the dark period with no change in the duration of NREM episodes (Fig 6I and J). Intriguingly, the number of sleep attempts during SD in old *Prdm13*-OE mice was significantly lower than control mice (Fig 6K). The level of SWA after SD in *Prdm13*-OE mice did not differ from old control mice (Figs 6L and S6J). Thus, the restoration of Prdm13 signaling in the DMH partially rescue age-associated sleep alterations, but its effect on sleep fragmentation is moderate.

### Prdm13 functions as a transcription factor in the DMH

What is the molecular function of Prdm13 in the DMH? We found that the DMH expresses previously uncharacterized alternative splicing variants of *Prdm13* by 5′-rapid amplification of cDNA-ends (RACE)-PCR analysis (Fig S7A and B). To further characterize the function of hypothalamic Prdm13 (htPrdm13), we developed an antibody against this variant. This antibody specifically detected the recombinant and overexpressed htPrdm13 proteins (data not shown) and also the deletion of Prdm13 in the DMH-*Prdm13*-KO mice (Fig 7A), confirming its specificity. Using this antibody, we examined the subcellular localization of the Prdm13 protein by biochemical fractionation (Fig 7B left). The Prdm13 protein was found exclusively in the RNase- and DNase-resistant nuclear scaffold fraction from WT hypothalami (Fig 7B right). Nuclear localization of Prdm13 in the hypothalamus was also confirmed by using hypothalami from a newly developed mouse model expressing podoplanin (PA)-tagged Prdm13 (Fig 7C). These results indicate that Prdm13 likely functions as a transcription factor in the DMH. This is consistent with a previous report showing that Prdm13, reported as Prdm13-202, acts as a transcription factor in the dorsal neural tube (Chang et al, 2013). Intriguingly, among DMH-enriched genes that were previously reported (Satoh et al, 2015), the levels of *cholecystokinin (Cck)*, *gastrin releasing peptide (Grp)*, and *pro-melanin–concentrating hormone (Pmch)* mRNA were significantly reduced in the compact region of the DMH (Bregma −1.79 and −1.91) of DMH-*Prdm13*-KO mice (Fig 7D). Transcriptional activities of *Cck* (one-way ANOVA, $F_{(1.016, 3.047)}$ = 10.89, $P$ = 0.0446), *Grp* (one-way ANOVA, $F_{(2.158, 6.474)}$ = 11.63, $P$ = 0.0068), and *Pmch* (one-way ANOVA, $F_{(1.547, 4.641)}$ = 54.50, $P$ = 0.0007) promoters were up-regulated by Prdm13-202 in a dose-dependent manner (Fig 7E), and Prdm13-202, htPrdm13 also up-regulated *Cck* transcription (one-way ANOVA, $F_{(1.010, 2.020)}$ = 49.88, $P$ = 0.0189) (Fig 7F). Moreover, a Prdm13 zinc finger (Zif) mutant with amino acid mutations (C187A, H207A, C622A, H638A,

hoc test (left) or unpaired *t* test (right). **(E)** Number of sleep attempts during SD from 6 ᴀᴍ to 8 ᴀᴍ (6–8), 8 ᴀᴍ to 9 ᴀᴍ (8–9), 9 ᴀᴍ to 10 ᴀᴍ (9–10), 10 ᴀᴍ to 11 ᴀᴍ (10–11), and 11 ᴀᴍ to 12 ᴘᴍ (11–12) in Prdm13-KO and Cont mice (n = 13–14). Values are shown as means ± S.E., *$P$ < 0.05, ***$P$ < 0.001 by repeated measures ANOVA with Bonferroni's post hoc test. **(F)** SWA during NREM sleep after SD. Normalized power is relative to the average of the 24-h baseline day each group (n = 6). Values are shown as means ± S.E. **(G)** EEG spectra of wakefulness (left), NREM sleep (middle), and REM sleep (right) during the light period (n = 5–6). Values are shown as means ± S.E. **(H)** Total amount of wakefulness, NREM sleep, and REM sleep during a 24-h period (24 h total), 12-h light period (12 h light) or 12-h dark period (12 h dark) (n = 6). Values are shown as means ± S.E.

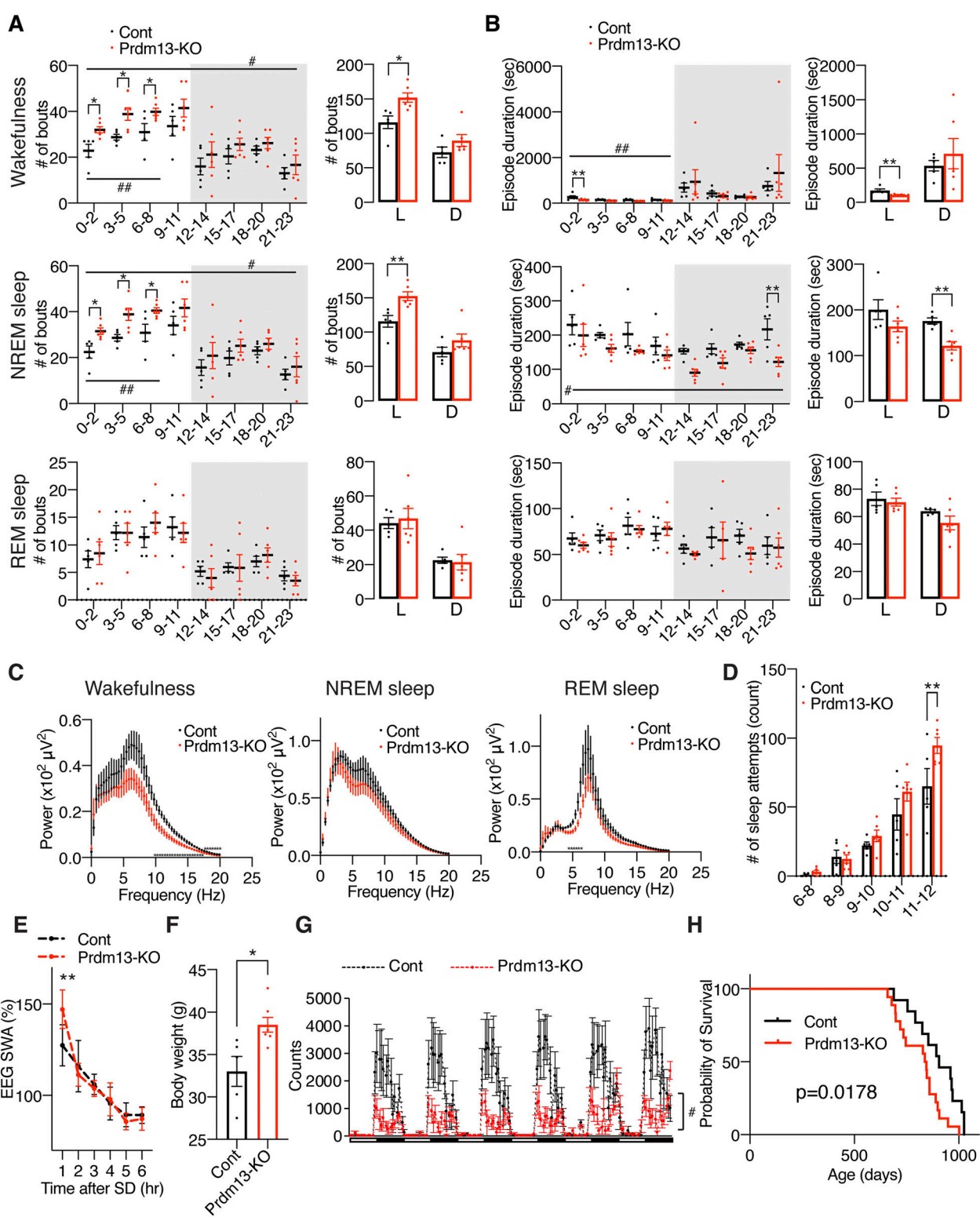

**Figure 5. Old DMH-*Prdm13*-KO mice display age-associated pathophysiology and shortened lifespan.**
**(A, B)** Numbers of episodes (A) and duration (B) of wakefulness (top), NREM sleep (middle), and REM sleep (bottom) every 3 h through a day (left) and during the light (L) and dark (D) periods (right) in old DMH-specific *Prdm13*-knockout (Prdm13-KO) and control (Cont) mice (n = 5–6). Values are shown as means ± S.E., #*P* < 0.05 and ##*P* < 0.01 by repeated measures ANOVA, *\*P* < 0.05 and \*\**P* < 0.01 by repeated measures ANOVA with Bonferroni's post hoc test (left) or unpaired *t* test (right). **(C)** EEG spectra of wakefulness (left), NREM sleep (middle), and REM sleep (right) during the light period (n = 4–6). Values are shown as means ± S.E., *\*P* < 0.01 by unpaired *t* test. **(D)** Number of sleep attempts during SD from 6 ᴀᴍ to 8 ᴀᴍ (6–8), 8 ᴀᴍ to 9 ᴀᴍ (8–9), 9 ᴀᴍ to 10 ᴀᴍ (9–10), 10 ᴀᴍ to 11 ᴀᴍ (10–11), and 11 ᴀᴍ to 12 ᴘᴍ (11–12) in old Prdm13-KO and Cont mice (n = 5–6). Values are shown as means ± S.E., \*\**P* < 0.01 by repeated measures ANOVA with Bonferroni's post hoc test. **(E)** SWA after SD of Prdm13-KO and Cont mice at 20 mo of

C650A, H666A, C679A, H695A), leading to inactivation of four Zif domains, showed significantly decreased transcriptional activity, whereas a Prdm13-PR/SET deletion mutant still activated these promoters to levels similar to Prdm13-202 (Fig 7E). These results reveal that the Zif domain, but not the PR/SET domain, is necessary for Prdm13 to up-regulate the activity of *Cck*, *Grp*, and *Pmch* promoters.

### Prdm13+Cck+ DMH neurons were activated during SD

Notably, at bregma −1.67 mm in the DMH, *Prdm13* was co-localized with *Cck* or *Grp* within the *Prdm13+* neuronal population about 26% and 14%, respectively (Fig 8A and B), but showed almost no co-localization with *Pmch* (Fig S8A). The percentage of *Prdm13+Cck+* in *Prdm13+* DMH neurons was significantly higher than the percentage of *Prdm13+Grp+* neurons (Fig 8B). Therefore, *Prdm13* might functionally or mechanistically connect with *Cck* and/or *Grp* in the DMH. *Cck+* cells among *Prdm13+* DMH neurons, particularly at bregma −1.67 mm, were widely distributed, but significantly more predominant in the medial part than the lateral part (16.1% ± 1.3% and 9.8% ± 1.3% in the medial and lateral parts, respectively, unpaired *t* test: $P < 0.001$), whereas *Grp+* cells among *Prdm13+* DMH neurons were distributed mainly in the lateral part (2.9% ± 1.1% and 10.9% ± 1.3% in the medial and lateral parts, respectively, unpaired *t* test: $P < 0.001$) (Fig 8A and B). During SD, 58% of *Prdm13+cFos+* neurons in the DMH were localized in the medial part (Fig 8C and D), where *Prdm13+Cck+* neurons were their majority, whereas only a few *Prdm13+Grp+* neurons were observed (Fig S8B). Thus, we questioned whether Cck is involved in the response to SD in this particular area. The percentage of *cFos+* cells within *Prdm13+Cck+* DMH neuronal population during SD was significantly higher than those in SD-Cont in young mice but not *Prdm13+Cck−* DMH neuronal population (Fig 8E). Thus, the specific neuronal population expressing both Prdm13 and Cck is activated in response to SD. Unexpectedly, the percentage of *cFos+* cells within *Prdm13+Cck+* DMH neuronal population during SD in old mice was also significantly higher than those in SD-Cont (Fig 8F). However, the degree of increase in *cFos+* in response to SD in young mice (2.0-fold) noticeably dropped in old mice (1.5-fold). Furthermore, the level of *Cck* in the hypothalamus of old mice was significantly lower compared with young mice (Fig S8C), and the number of *Prdm13+* DMH cells that highly expressed Cck in old mice tended to be lower than young mice (Fig S8D). Therefore, decreased level of *Cck* expression in the DMH might affect sleep–wake patterns in old mice, and these effects might occur with affecting neuronal activation and another neuronal event.

## Discussion

We demonstrated that Prdm13 signaling in the DMH is responsible for the remarkable effect of DR against age-associated sleep fragmentation and excessive sleepiness during SD in mice.

Consistently, in humans, 2 yr of 25% DR promoted better sleep quality reflected by lower scores reported on the Pittsburgh Sleep Quality Index compared with the AL group (Martin et al, 2016). Another study showed that 2 d of 10% DR significantly increased the duration of stage 4 deep NREM sleep (Collet et al, 2016). Our study revealed that the deficiency of Prdm13 signaling promotes some phenotypes similar to age-associated sleep–wake alterations. Thus, maintaining Prdm13 signaling in the DMH might be beneficial not only to prevent age-associated sleep alterations but also to ameliorate age-associated pathophysiologies in old animals. Further elucidating detailed downstream molecular events regulated by Prdm13 signaling and characteristics of Prdm13+ DMH neurons will be of great interest to explore a potential intervention on age-associated sleep–wake patterns.

A potential mechanism by which Prdm13 signaling in the DMH controls sleep fragmentation and excessive sleepiness during SD is the transcriptional regulation of neuropeptides in the DMH. In particular, *Cck* transcription is down-regulated within the hypothalamus of old mice. Optogenetic and chemogenetic studies show that activation of GABAergic/Cck+ neurons in the preoptic area of the hypothalamus promotes NREM sleep (Chung et al, 2017). Similarly, the activation of glutamatergic/Cck+ neurons in the perioculomotor region of the midbrain also promotes NREM sleep likely through the activation of GABAergic neurons in the preoptic area of the hypothalamus. Together, it is likely that activation of Cck+ neurons in the preoptic area of the hypothalamus and the perioculomotor region of the midbrain promotes sleep. Young DMH-*Prdm13*-KO mice display excessive sleepiness during SD to the extent equivalent to old mice. Because sleep fragmentation is further developed with age, it is conceivable that the decreased function of Prdm13 signaling causes excessive sleepiness during SD, and then sleep fragmentation. The detailed mechanisms by which the deficiency of *Prdm13* leads to sleep fragmentation still need to be elucidated. For instance, it would be of great interest to elucidate whether Prdm13 signaling in the DMH contributes to regulate the circadian system because the DMH is known to be involved in the regulation of several circadian behaviors (Aston-Jones et al, 2001; Chou et al, 2003). Although DMH-*Prdm13*-KO mice did not display abnormal period length compared with controls, further studies are needed to address this possibility.

At young age, DMH-*Prdm13*-KO mice display sleep fragmentation and excessive sleepiness during SD, mimicking sleep changes observed in old WT mice. In addition, DMH-*Prdm13*-KO mice show exacerbated sleep fragmentation and develop other physiological changes such as increased adiposity and decreased wheel-running activity over age. The reason for DMH-*Prdm13*-KO mice to show changes primarily in sleep, and secondarily in body weight and physical activity at older age is currently unknown. One possibility is that prolonged sleep fragmentation induces low physical activity and increased adiposity. To support this idea, it has been reported that 20 d of sleep fragmentation during the light period promotes a

---

age. Normalized power is relative to the average of the 24-h baseline day (n = 5–6). Values are shown as means ± S.E., **$P < 0.01$ by Bonferroni's post hoc test. **(F)** Body weight of old Prdm13-KO and Cont mice (n = 5–7). Values are shown as means ± S.E., *$P < 0.05$ by unpaired *t* test. **(G)** The level of wheel-running activity in old Prdm13-KO and Cont mice for six consecutive days (n = 5–7). Values are shown as means ± S.E., #$P < 0.05$ by repeated measures ANOVA. **(H)** Kaplan–Meier curves of Prdm13-KO and Cont mice (n = 13–18). Listed *P*-value was calculated by log-rank test.

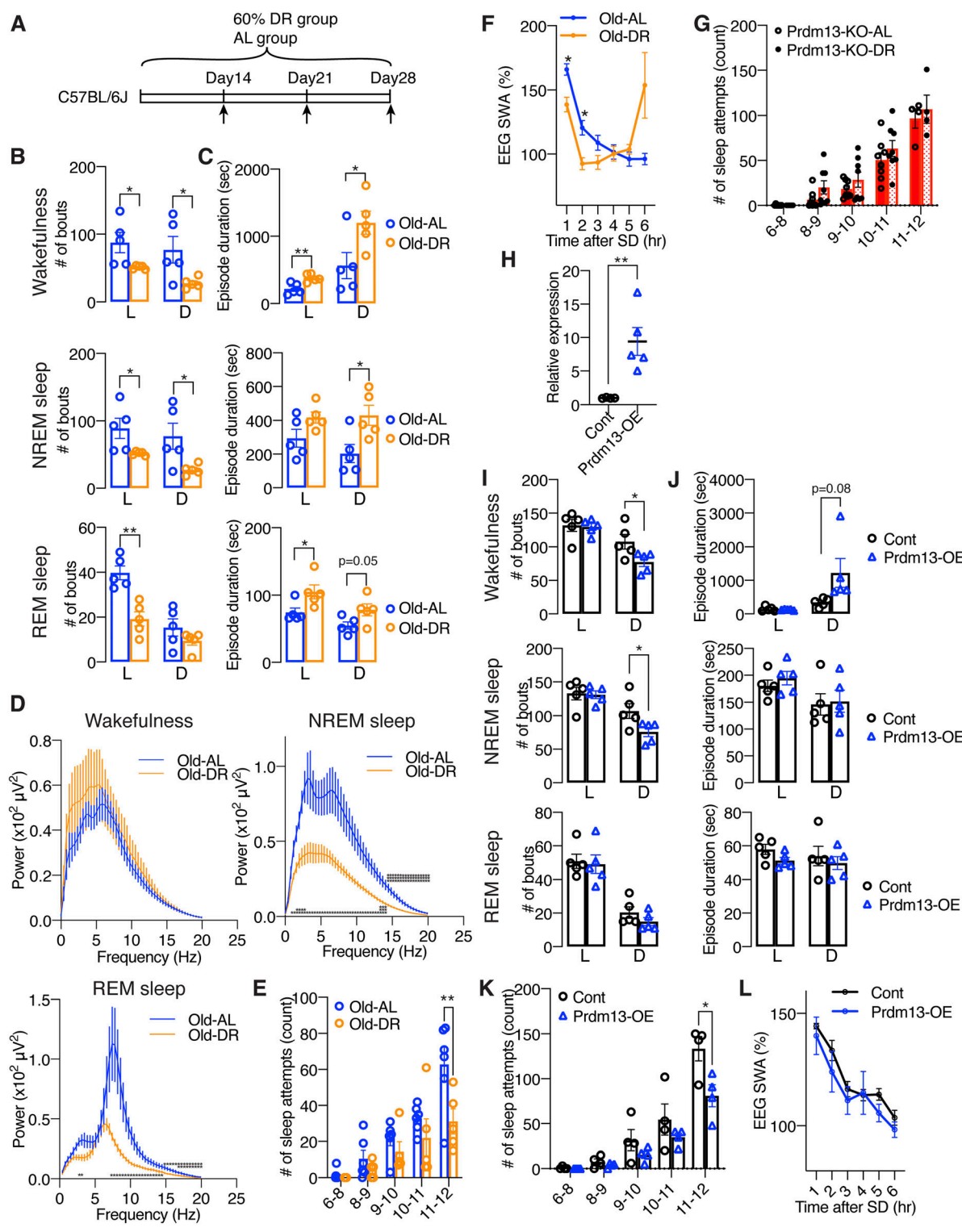

**Figure 6. DR and overexpression of *Prdm13* in the DMH ameliorates age-associated sleep fragmentation and excessive sleepiness during SD.**
**(A)** DR paradigm in C57BL/6J at 20 mo of age. Mice at 20-mo-old were fed under 60% diet or AL-diet for 14–28 d. **(B, C)** Number of episodes (B) and duration (C) of wakefulness (top), NREM sleep (middle), and REM sleep (bottom) during the light (L) and dark (D) periods in AL and DR mice at 20 mo of age (n = 5). Values are shown as means ± S.E., listed *P*-value, *$P < 0.05$ and **$P < 0.01$ by unpaired *t* test. **(D)** EEG spectra of wakefulness (upper left), NREM sleep (upper right), and REM sleep (lower) during the light period (n = 5). Values are shown as means ± S.E., *$P < 0.05$, **$P < 0.01$, ***$P < 0.001$ by unpaired *t* test. **(E)** Number of sleep attempts during SD from 6 AM to 8 AM (6–8), 8 AM to 9 AM (8–9), 9 AM to 10 AM (9–10), 10 AM to 11 AM (10–11), and 11 AM to 12 PM (11–12) in AL and DR mice at 20 mo of age (n = 5–6). Values are shown as means ± S.E., **$P < 0.01$ by repeated measures ANOVA with Bonferroni's post hoc test. **(F)** SWA after SD of AL and DR mice at 20 mo of age. Normalized power is relative to the average of the 24-h baseline day (n = 5). Values are shown as means ± S.E., *$P < 0.05$ by Bonferroni's post hoc test. **(G)** Number of sleep attempts during SD from 6 AM to 8 AM (6–8),

decreased physical activity in young mice (He et al, 2015). In humans, low sleep efficiency, a widely recognized index of sleep consolidation and fragmentation, is significantly associated with the reduction in daytime physical activity (Lambiase et al, 2013; Kline, 2014). Therefore, the low physical activity observed in old DMH-*Prdm13*-KO mice might be a consequence of chronic sleep fragmentation. Similar to physical activity, obesity is also promoted by sleep fragmentation through an increased food intake in mice (Wang et al, 2014; Hakim et al, 2015), but old DMH-*Prdm13*-KO mice do not alter their food intake. Therefore, the increased adiposity in old DMH-*Prdm13*-KO mice might be a consequence of low physical activity.

In this study, we examined age-associated changes in sleep–wake patterns, EEG spectra, and responsiveness to SD. Consistent with previous reports (Wimmer et al, 2013; Panagiotou et al, 2017; McKillop et al, 2018; Soltani et al, 2019), we confirmed that old mice showed reduced total amount of wakefulness and increased NREM sleep during a 24-h period. Such differences are more pronounced during the dark period. Nonetheless, our findings in this study implicate an important possibility that old mice are more susceptible to accumulate sleep pressure compared with young mice. In fact, the level of SWA after SD, which reflects accumulated sleep pressure from increased wakefulness, in old mice was significantly higher than young mice. Indeed, our finding is consistent with a notion that old mice live under a high sleep pressure (Panagiotou et al, 2017). Because the homeostatic sleep response is fairly intact with age, accumulation and generation of sleep pressure in old mice might be greater than that in young mice. However, there are clear discrepancies between mice and humans in sleep studies (Campos-Beltran & Marshall, 2021). It should be noted that some of the age-associated sleep alterations such as sleep fragmentation are conserved in both mice and human, whereas basic sleep architecture is quite different between each other (e.g., polyphasic versus monophasic sleep, nocturnal versus diurnal). In most reports, aged mice show an increase in total NREM sleep in the dark period (Wimmer et al, 2013; Panagiotou et al, 2017; McKillop et al, 2018; Soltani et al, 2019), but older humans show a decrease (Panagiotou et al, 2017; McKillop et al, 2018). In this regard, it is interesting that a most recent study has reported that a recessive mutation to the human PRDM13 gene causes ataxia with cerebellar hypoplasia and delayed puberty with hypogonadotropic hypogonadism (Whittaker et al, 2021). Recent GWAS study revealed that PRDM13 is listed as one of the potential nocturnal enuresis risk genes (Jorgensen et al, 2021). It remains unclear whether these patients also have sleep defects. Thus, although mice are useful for modeling certain aspects of aging and sleep, an extra caution will be necessary in extrapolating the results obtained from mice to humans.

We discovered Prdm13 signaling in the DMH affects some sleep–wake patterns during the aging process. We also uncovered that Prdm13 signaling is necessary to promote DR effects in age-

associated sleep patterns and restoration of Prdm13 in the DMH partially affect age-associated sleep alterations. Chemogenetic manipulation of Prdm13+ neurons in the DMH reveals that the inhibition of this neuronal population promotes excessive sleepiness during SD, demonstrating the direct involvement of Prdm13+ DMH neurons in controlling sleep–wake patterns through neuronal activity. Our study elucidated that Prdm13 acts as a transcription factor that regulates the expression of critical neuropeptides in the DMH. Among those neuropeptides, *Cck* is likely involved in the regulation of age-associated sleep–wake pattern changes mediated by Prdm13 functioning as a transcription factor in the DMH. Other downstream target genes of Prdm13 may also potentially be involved in the age-associated regulation of sleep–wake patterns. The detailed relationship between the activity of Prdm13+ DMH neurons and Prdm13 signaling itself still need to be elucidated in the future studies.

# Materials and Methods

### Animal models

All mouse experiments and procedures were approved by the Animal Care and Use of the NCGG. Mice were housed in 12/12-h light/dark cycle (lights on at 6 AM and off at 6 PM) with free access to food and water. 1–2 mo-old C57BL/6J mice were purchased from the Charles River Laboratories International, Inc. and grew up to 4 or 20–22 mo of age (as young and old groups, respectively) in our Animal Facility at the NCGG and aged mice specialized suits at the NCGG. *Rosa26R*$^{Zs-Green/ZsGreen}$ and *Nkx2-1*$^{CreERT2/+}$ mice (Jackson stock no: 007906 and 014552) were obtained from Jackson Laboratory. *Prdm13*$^{fl/fl}$ mice (RIKEN BRC stock no: RBRC09371) (Watanabe et al, 2015) were obtained from RIKEN BRC. Most of the mice were housed in groups, except for the DR study. For the DR study, C57BL/6J mice at 20 or 4 mo of age or DMH-*Prdm13*-KO mice at 12 mo of age were fed under 60% diet or AL-diet for 28 d. To minimize habitual stress and disruption of daily pattern, food was gradually decreased to 60% of daily food intake, and both AL and DR groups were fed daily at 5–6 PM right before the dark period. Mice were closely monitored and included daily body weight measurement during the experimental period. Female mice were mainly used for sleep studies, wheel-running analysis, food intake behavior studies, and DR studies. Both male and female mice were used to confirm excessive sleepiness during SD in DMH-*Prdm13*-KO mice. Only male mice were used in longevity study and the DR study using DMH-*Prdm13*-KO mice.

*Prdm13*-CreERT2 mice were generated by the Laboratory Animal Resource Center at the University of Tsukuba. The detailed

---

8 AM to 9 AM (8–9), 9 AM to 10 AM (9–10), 10 AM to 11 AM (10–11), and 11 AM to 12 PM (11–12) in Prdm13-KO-AL and Prdm13-KO-DR mice (n = 8). Values are shown as means ± S.E. **(H)** Expression of *Prdm13* in the hypothalamus of *Prdm13*-overexpressing (Prdm13-OE) and control (Cont) mice (n = 4–5). Values are shown as means ± S.E., *$P$ < 0.05 by unpaired $t$ test. **(I, J)** Number of episodes (I) and duration (J) of wakefulness (top), NREM sleep (middle), and REM sleep (bottom) during the light (L) and dark (D) periods in Prdm13-OE and Cont mice (n = 5). Values are shown as means ± S.E., *$P$ < 0.05 by unpaired $t$ test. **(K)** Number of sleep attempts during SD from 6 AM to 8 AM (6–8), 8 AM to 9 AM (8–9), 9 AM to 10 AM (9–10), 10 AM to 11 AM (10–11), and 11 AM to 12 PM (11–12) in Prdm13-OE and Cont mice (n = 4). Values are shown as means ± S.E., *$P$ < 0.05 by repeated measures ANOVA with Bonferroni's post hoc test. **(L)** SWA after SD of Prdm13-OE and Cont mice. Normalized power is relative to the average of the 24-h baseline day (n = 4). Values are shown as means ± S.E.

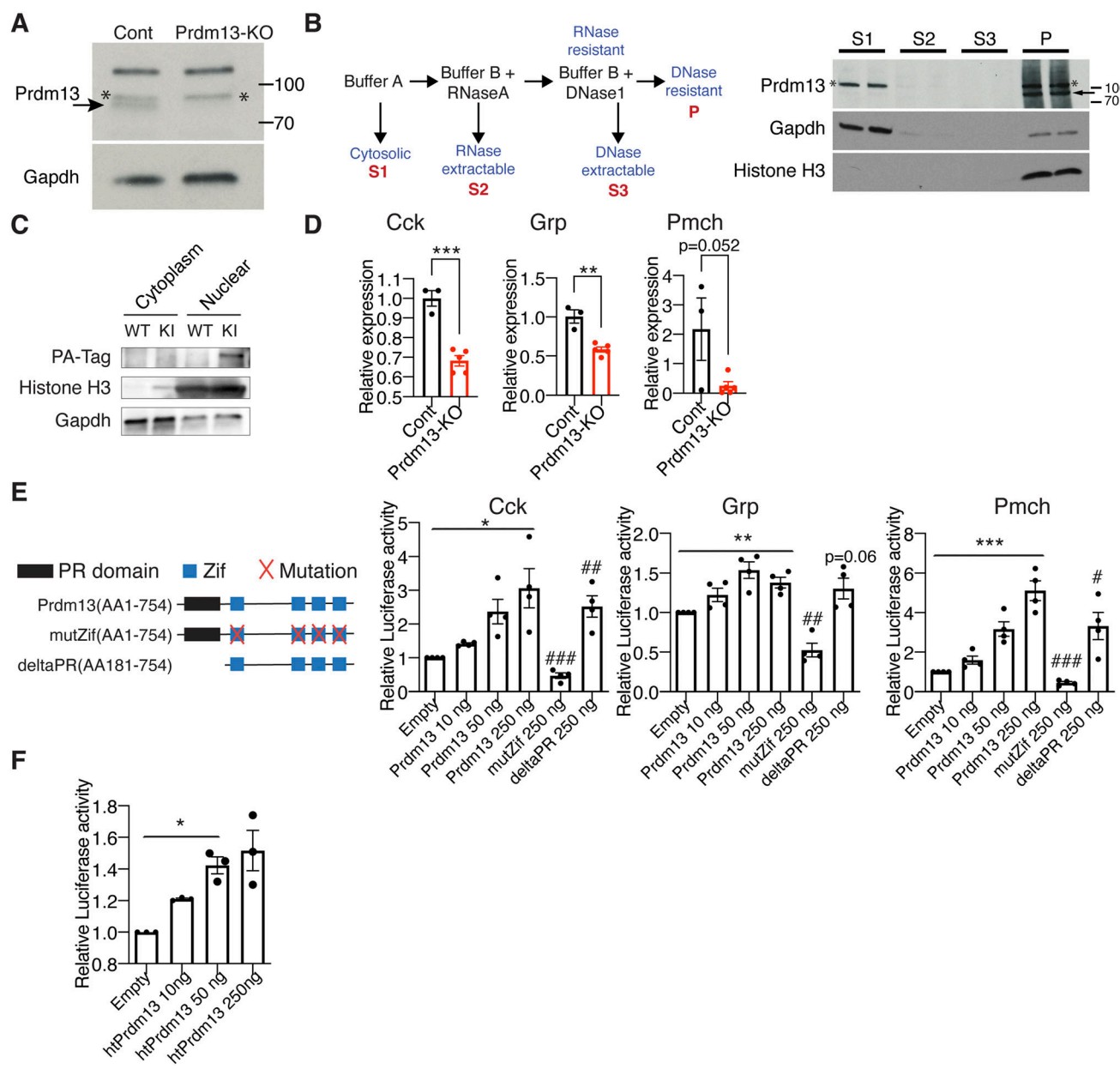

**Figure 7. Prdm13 in the DMH is a transcription factor.**
**(A)** Western blot of Prdm13 in DMH collected by laser microdissection from DMH-specific Prdm13-KO and Cont mice (n = 4 mice/lane). The arrow indicates the band for Prdm13; asterisks (*) indicate non-specific bands. **(B)** Schematic of fractionation protocol from mouse hypothalami (left). Western blot of Prdm13 in hypothalamic fractions of C57BL/6J mice (right). Hypothalami from two C57BL/6J female mice were combined for each lane, and 8% equivalent of each fraction was run on the gel. Cytoplasmic supernatant (S1), RNase-extractable supernatant (S2), DNase-extractable supernatant (S3), and insoluble pellet (P) were run each lane. The arrow indicates the band for Prdm13; asterisks (*) indicate non-specific bands. **(C)** Western blot of Prdm13 in hypothalamic fractions of *Prdm13*-PA-Tag (KI) and WT mice. Cytosolic and nuclear fractions were run each lane as indicated. **(D)** Expression of *Cck*, *Grp*, and *Pmch* mRNA in the DMH of DMH-*Prdm13*-KO (Prdm13-KO) and control (Cont) mice (n = 3–5). Values are shown as means ± S.E., listed *P*-value, **P < 0.01 and ***P < 0.001 by unpaired *t* test. **(E)** Transcriptional activity of Prdm13-202 and Prdm13-mutants for the luciferase reporter vector containing the promoter region of *Cck*, *Grp*, and *Pmch*. Schematic representation of Prdm13-202 and Prdm13-mutants are shown above. NIH3T3 cells were co-transfected with 250 ng of luciferase reporter plasmid and plasmid expressing Prdm13-202 (Prdm13), Prdm13-Zif mutant (mutZif), or Prdm13-deltaPR mutant (deltaPR). Obtained luminescence was normalized to total protein concentration (n = 3, four individual experiments). Values are shown as means ± S.E., listed *P*-value, *P < 0.05, **P < 0.01, and ***P < 0.001 by one-way ANOVA with Bonferroni's post hoc test, #P < 0.05 and ##P < 0.01 and ###P < 0.001 by unpaired *t* test. **(F)** Transcriptional activity of hypothalamic Prdm13 (htPrdm13) for the luciferase reporter plasmid containing the promoter region of *Cck*. NIH3T3 cells were co-transfected with 250 ng of reporter plasmid and 10, 50, or 250 ng of *htPrdm13*-expressing plasmid. Obtained luminescence was normalized to total protein concentrations (n = 3, three individual experiments). Values are shown as means ± S.E., *P < 0.05 by one-way ANOVA with Bonferroni's post hoc test.

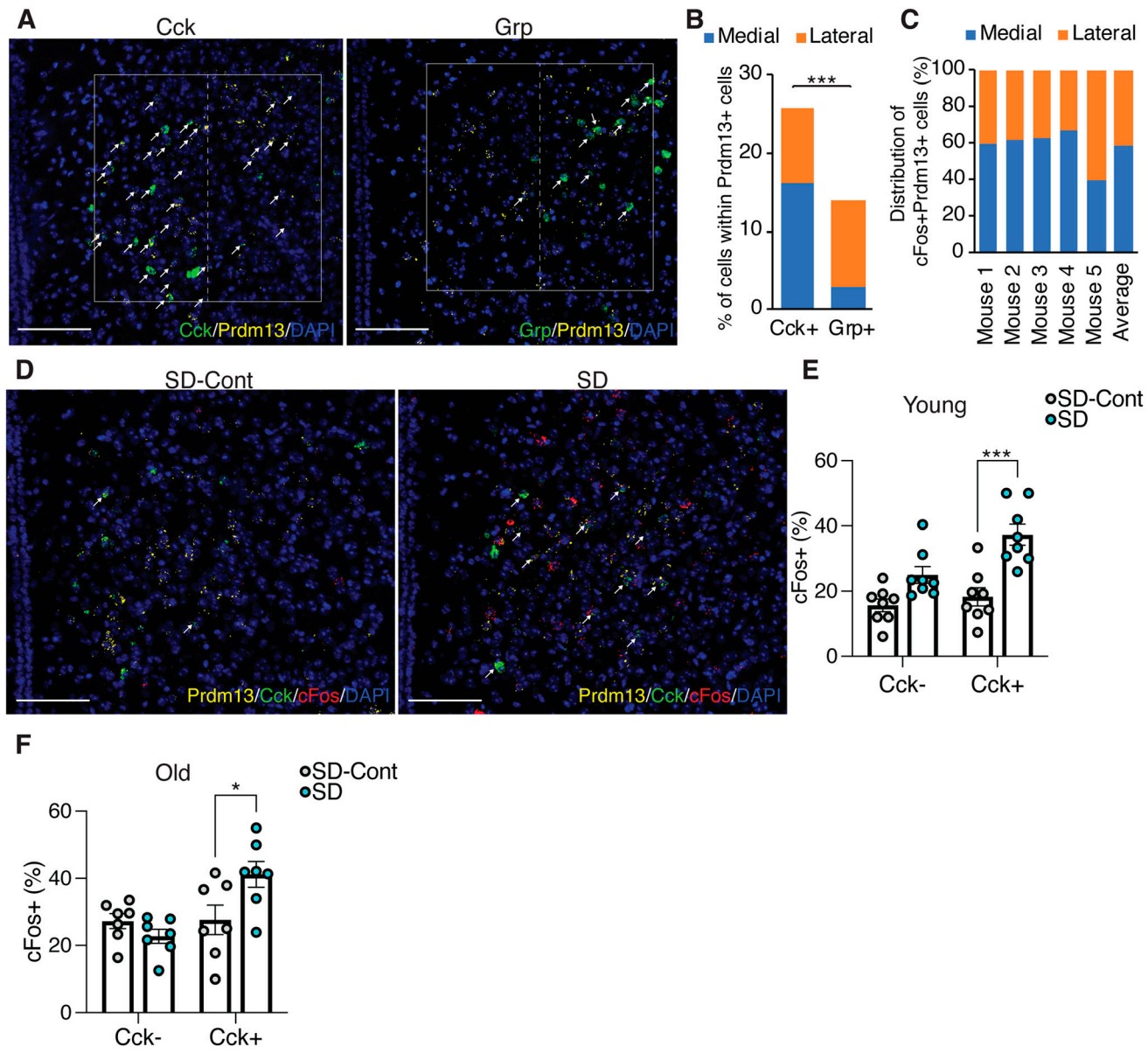

**Figure 8. DMH-Prdm13+Cck+ neurons are activated during SD.**
**(A)** Representative images of the DMH with *Prdm13* (yellow) and one of the two genes, *Cck* or *Grp* (green) visualized by RNAscope. Cells were counterstained with DAPI (blue). White boxes show the DMH, which is divided into medial and lateral areas by dashed lines. White arrows show yellow+green+ cells. Scale bar indicates 100 μm. **(B)** Ratios of *Cck*+ or *Grp*+ cells within *Prdm13*+ cells in medial, lateral, or total (medial and lateral) DMH (n = 3–5). Values are shown as means ± S.E., ***P < 0.001 by unpaired *t* test. **(C)** Distribution of *cFos*+*Prdm13*+ cells (n = 5). **(D)** Representative images of the DMH from young mice under SD-Cont and SD with *Prdm13* (yellow), *Cck* (green), and *cFos* (red) visualized by RNAscope. Cells were counterstained with DAPI (blue). White arrows show *Prdm13*+*Cck*+*cFos*+ cells. Scale bar indicates 100 μm. **(E, F)** Ratios of *cFos*+ cells within *Prdm13*+*Cck*– (left) or *Prdm13*+*Cck*+ (right) cells in young (E) or old (F) mice during SD-Cont and SD (n = 7–8). Values are shown as means ± S.E., *P < 0.05, ***P < 0.001 by two-way ANOVA with Bonferroni's post hoc test.

procedure was described previously (Mizuno-Iijima et al, 2021). Briefly, a targeting vector was designed to insert the *Prdm13* sgRNAs (5′-GACTCCTAACGCGCCTTCCA-3′) into pX330-mC plasmid, which carried Cas9-mC expression unit (Mizuno-Iijima et al, 2021) (pX330-mC-Prdm13sgRNA). pCreERT2-Prdm13 was designed to insert a 2A peptide (P2A), CreER[T2] recombinase, and rabbit globin polyadenylation signal, replacing TAA stop codon in the fourth exon of *Prdm13* gene. These two constructs (pX330-mC-

Prdm13sgRNA and pCreERT2-Prdm13) were microinjected into zygotes from C57BL/6J mice. Subsequently, injected zygotes were transferred into oviducts in pseudopregnant ICR female mice (Charles River Laboratories International, Inc.), and 85 newborns were obtained. The designed knock-in (KI) mutation was confirmed by PCR using the following primers: Prdm13 screening 5Fw: 5′-CATGCACAGCACTTGTGGTAGAGAAATC-3′, Prdm13 screening 3Rv: 5′-ATTTAGAATTGGAGCAAACAGGGGGATT-3′. No random integrations

were detected by PCR with primers detecting the ampicillin resistance gene.

*Prdm13*-PA-Tag KI mice were generated by the Laboratory Animal Resource Center at the University of Tsukuba. We attempted to introduce the PA-Tag coding sequence connected to the LG3-linker sequence just before the TAA stop codon of *Prdm13* gene (Kagoshima et al, 2007). Briefly, the gRNA (5′-AGTCCCTG-GAAGGCGCGTT-3′) was synthesized and purified by GeneArt Precision gRNA Synthesis Kit (Thermo Fisher Scientific). In addition, we designed a 200-nt single-stranded DNA oligonucleotide donor, placing the LG3-PA sequence between the genomic regions from 54 bp upstream of the TAA stop codon to 53 bp downstream of the TAA (Integrated DNA Technologies). The gRNA, single-stranded DNA oligonucleotide, and GeneArt Platinum Cas9 Nuclease (Thermo Fisher Scientific) were electroporated to C57BL/6J zygotes using a NEPA 21 electroporator (NEPAGNENE), as described previously (Sato et al, 2018). After electroporation, two-cell embryos were transferred into oviducts in pseudopregnant ICR female mice, and 32 newborns were obtained. The PA-Tag KI mutation was confirmed by PCR using the following primers: Prdm13 QC primer F: 5′-TCAA-CAAGCACATCCGACTC-3′, Prdm13 QC primer R: 5′-TGACGTGATCCT-GAACCTCA-3′. The PCR products were sequenced by using BigDye Terminator v3.1 Cycle Sequencing Kit (Thermo Fisher Scientific).

## Sleep analysis

Isoflurane-anesthetized mice were surgically implanted with stainless screw electrodes placed over the right frontal bone for reference and right/left parietal bone for active recording EEG, and wire electrodes in the nuchal muscle for EMG recording. All signals were grounded to a bone screw electrode placed over the cerebellum midline. Mice were recovered from surgery for 3 d and subsequently acclimatized to the recording cage for 3 wk. EEG/EMG recording was performed continuously for two consecutive days. Recording electrodes were connected to a TBSI Tethered System T8 amplifier (TBSI) via T8 Headstage (A50-2139-G3, TBSI), a lightweight cable and commutator to enable free movement and feeding in a sound and light proof enclosure with a 12/12-h light/dark cycle. EEG/EMG signals were digitized at 600 Hz, filtered at 0.3–35 Hz for EEG, and 10–100 Hz for EMG by PowerLab system (ADInstruments). Wireless EEG Logger (ELG-2; Bio research Center) was used for sleep analysis in Fig 6I–L. 10-s epochs of EEG/EMG signals were semi-automatically scored as wakefulness, NREM sleep, and REM sleep by SleepSign (KISSEI COMTEC) with visual examination. EEG periods dominated by higher amplitude delta wave activity with nuchal muscle atonia were scored as NREM sleep epochs. REM sleep consisted of periods of semi-uniform theta activity EEG with muscle atonia and/or muscle atonia with brief myoclonic twitches. Score was blinded for genotypes during quantification. Spectra analysis was performed by an FFT (FFT; 0.4–20 Hz, 0.38 Hz resolution). Two outliers were detected using Grubb's and ROFU tests (GraphPad Prism 9) (one KO mouse in Figs 4 and 5) and excluded from spectrum analysis. SWA during NREM sleep was computed across the 24-h recording period by SleepSign (KISSEI COMTEC). SWA after SD was normalized to the average of SWA for the average of 24-h period each mouse.

## SD study

Mice were individually housed before the experiment. On the day of SD, food was removed at 6 am and mice were kept awake using a long Q-tip until 12 pm (6 h SD) by gentle touching of mice, as previously reported (Franken et al, 1991). Attempts to sleep were determined by the onset of behaviors typical of sleep such as cornering, curling, and eye closing. Once such behaviors were observed, we placed the long Q-tip in front of the mouse. One sleep attempt was counted when the mouse did not react to it. EEG/EMG recording was performed during SD to monitor the effectiveness of the SD protocol. All animals indicated <5% of sleep during the 6 h of SD. After SD, food was added, and the mice were allowed to sleep. Mice for control manipulation (AL-SD) were also individually housed before the experiment and during the experiment food was removed as in the SD group. We checked the level of blood glucose in the SD study and found that the level of blood glucose was indistinguishable between SD and AL-SD groups (126 ± 6 and 131 ± 4 mg/dl, respectively), revealing that nutritional status is equal between these two groups. Genotypes and conditions were blinded during the experimental procedures.

## Immunohistochemistry and immunofluorescence

Mice were anesthetized with isoflurane and perfused with PBS followed by 4% PFA at 11 AM for SD and SD-Cont, and at 2 PM for RS and RS-Cont. Brains were fixed with 4% PFA overnight and placed into 30% sucrose until saturated. Thirty-micrometer cryosections were collected into PBS and stored in cryoprotectant at −20°C. For immunofluorescent staining, samples were stained using primary antibodies: anti-Nkx2-1 (TTF-1) (1:500, ab76013; Abcam) and secondary antibodies. To stain cFos, samples were stained with anti-cFos (1:1,000, 226003; Synaptic Systems) and universal biotinylated anti-mouse/rabbit IgG (Universal Elite ABC kit, PK-7200; Vector laboratories) antibodies with Universal Elite ABC kit and developed with Vector SG substrate kit peroxidase (SK-4700: Vector laboratories). The number of cFos-positive cells was quantified by visual scoring. Genotypes and conditions were blinded during the experimental procedures. For MAP2 immunofluorescence, 25-μm cryosections were stained using anti-MAP2 (1:100, ab5392; Abcam).

## In situ hybridization

For RNAscope, brains from C57BL/6J mice were dissected, embedded in OCT, and frozen on dry ice. The embedded frozen blocks were cut at 14 μm thick using cryostat CM1850 (Leica) and mounted on slides. The sections were stored at −80°C until further processing. Target mRNA was detected using the RNAscope Multiplex Fluorescent Reagent Kit v2 (Advanced Cell Diagnostics [ACD]). RNA probes (ACD) used in this study are as follows: *Prdm13* (Cat# 543551-C2), *Fos* (Cat# 316921-C3), *Cck* (Cat# 402271), *Grp* (Cat# 317861), and *Pmch* (Cat# 478721). The frozen sections were fixed in pre-chilled 4% PFA in PBS for 10 min. After 2 times washing with PBS, the sections were dehydrated through 50%, 70%, 100%, and 100% ethanol for 5 min each. The slides were air dried for 5 min. The slides were treated with hydrogen peroxide for 10 min at room temperature. Probe hybridization and signal amplification were performed using TSA Plus kit (PerkinElmer) according to the ACD's instructions. The

slides were counterstained with DAPI and mounted using 2.5% 1,4,diazabicyclo[2.2.2]octane (DABCO) in 50% glycerol. The slides were imaged with LSM700 laser-scanning confocal microscope (Zeiss) with ZEN 2009 software (Zeiss). Cells positive for *Prdm13*, *cFos*, *Cck*, *Grp*, and *Pmch* were manually detected. To evaluate *Cck* expression level semiquantitatively, signal dots derived from *Cck* mRNA were counted manually and categorized into four grades: 1 (1–5 dots/cell), 2 (6–10 dots/cell), 3 (11–15 dots/cell), and 4 (>16 dots/cell). Genotypes and conditions were blinded during the experimental procedures.

## Lentivirus production

To generate the *Prdm13*-expressing lentiviral construct, *Prdm13* cDNA was cloned into the FCIV.FM1 vector (a gift from the Viral Vectors Core at Washington University School of Medicine). High-tittered viruses were generated from the Viral Vectors Core at Washington University School of Medicine. Briefly, lentiviruses were produced by co-transfecting HEK293T cells with the *Prdm13*-expressing vectors and three packaging vectors (pMD-Lg, pCMV-G, and RSV-REV) by the calcium phosphate precipitation procedure. 6 h after transfection, the medium was replaced with the complete medium containing 6 mM sodium butyrate. Culture supernatant was collected 42 h after transfection. The supernatant was passed through a 0.45-$\mu$m filter, concentrated by ultracentrifugation through a 20% sucrose cushion, and stored at −80°C until use. Virus titer was determined by transducing HT1080 cells and assaying for reporter expression using flow cytometry.

## AAV and lentivirus injections

Following anesthesia with isoflurane gas, the mouse was placed in a three-point fixation stereotactic frame. Bregma was identified, and appropriate coordinates for the stereotactic injection were registered: relative to bregma for the DMH, anterior–posterior (AP) −1.4 mm, medial–lateral (ML) ± 0.3 mm, and dorsal–ventral (DV) −5.4 mm. A burr hole was made using a dental drill, and a glass capillary was directed to the previously determined coordinates. Viruses were slowly injected (100 nl/2 min). After the injection, animals were allowed to recover in a temperature-regulated incubator (32°C) until fully awake. All injected mice had 4 wk to fully recover before being used for any experiments. Viruses with the following titers and volumes were injected: AAV8-hSyn-DIO-nM4Di-mCherry (Addgene no 44362, $\geqq$ ×10$^{13}$ Vg/ml, 400 nl in the DMH). Lentiviruses carrying *Prdm13* or *fLuc* cDNA (2.0 × 10$^8$ IU/ml, 500 nl in the DMH).

## Chemogenetic modulation of Prdm13+ neurons during SD

4 wk after injecting AAV8-hSyn-DIO-nM4Di-mCherry, tamoxifen was intraperitoneally injected into mice for three consecutive days (200 mg/kg body weight/day). After 1 wk, *Prdm13*$^{CreERT2/+}$ or *Prdm13*$^{+/+}$ mice with Prdm13+ neurons expressing hSyn-DIO-hM4Di-mCherry were implanted with screw electrodes in the skull for EEG and EMG recording as described above. Mice were recovered from surgery for three days and subsequently acclimatized to the recording cage for 3 to 4 wk. Saline or CNO (1 mg/kg body weight) was intraperitoneally injected at 8 AM. Sleep was

deprived from 6 AM until 11 AM. During SD, sleep attempts were counted as described above. For basal measurement, saline or CNO was intraperitoneally injected at 11 AM.

## Whole-cell patch-clamp electrophysiology

Mice were anesthetized with an isoflurane–oxygen mixture, and the brain was removed. The brain was quickly transferred into ice-cold dissection buffer (25 mM NaHCO$_3$, 1.25 mM NaH$_2$PO$_4$, 2.5 mM KCl, 0.5 mM CaCl$_2$, 7 mM MgCl$_2$, 25 mM glucose, 110 mM choline chloride, 11.6 mM ascorbic acid, 3.1 mM pyruvic acid, and 1 mM kynurenic acid), gassed with 5%CO$_2$/95%O$_2$. Coronal brain slices were cut (300 $\mu$m; Leica VT1200S) in dissection buffer. The slices were then incubated in physiological solution (118 mM NaCl, 2.5 mM KCl, 26 mM NaHCO$_3$, 1 mM NaH$_2$PO$_4$, 10 mM glucose, 4 mM MgCl$_2$, 4 mM CaCl$_2$, pH 7.4, gassed with 5%CO$_2$/95%O$_2$).

Patch recording pipettes (3–7 M$\Omega$) were filled with intracellular solution (115 mM cesium methanesulfonate, 20 mM CsCl, 10 mM HEPES, 2.5 mM MgCl$_2$, 4 mM Na$_2$ATP, 0.4 mM Na$_3$GTP, 10 mM sodium phosphocreatine, and 0.6 mM EGTA at pH 7.25). To record the mEPSC (−60 mV holding potential) or mIPSC (0 mV holding potential), the recording chamber was perfused with physiological solution with 0.5 $\mu$M TTX. Whole-cell recordings were obtained from Prdm13+ neurons of the DMH with a MultiClamp 700B (Axon Instruments). Cells with membrane resistance >100 M$\Omega$ and series resistance <20 M$\Omega$ were only recorded. Whole-cell patch-clamp data were collected with Clampex and analyzed using Clampfit 10.7 software (Axon Instruments) (Tada et al, 2016).

## Identification of 5′-end of htPrdm13 cDNA from mouse hypothalami

We performed 5′-rapid amplification of cDNA-ends (5′-RACE)-PCR analyses to determine the 5′ end of htPrdm13 transcripts in RNAs isolated from the hypothalamus of C57BL/6J. The hypothalamus of C57BL/6J were dissected and immediately frozen in liquid nitrogen. Total RNA from the hypothalamus was extracted with RNeasy kit (QIAGEN). 5′-RACE was performed by SMARTer RACE 5′/3′ kit (Clontech) by following the manufacturer's protocol. Briefly, first-strand cDNA was synthesized with 5′ RACE CDE Primer A (Clontech) (5′-RACE-ready cDNA samples), and the 5′-RACE-ready cDNA sample was used for 5′-RACE reaction with a htPrdm13 specific primer (5′-GATTACGCCAAGCTT-TAGCGAAAGGTCCTCCAGCAGTA-3′). RACE products were purified by NucleoSpin Gel (QIAGEN) and PCR Clean-Up Kit (QIAGEN). Purified RACE products were inserted into pRACE vector with In-Fusion DM Master Mix. 5′-sequencing was confirmed by reading more than seven independent clones in 5′-RACE products using primer (5′-AAGCTTGGCGTAATC-3′). PCR was conducted using 5′-end primer (5′-ATGGTGAGAGGGGAGCTGGT-3′) and 3′-end primer (5′-TTAGGAGTCGTGCTCGCCAC-3′) to confirm amplification of htPrdm13 using hypothalamic cDNAs.

## Western blot analysis of Prdm13 from mouse hypothalamus

For antibody confirmation, the compact region of DMH from four DMH-*Prdm13*-KO or control mice was collected by laser microdissection

into Laemmli's sample buffer using the Leica LMD 6000 system (Leica) and boiled 5 min. The detailed procedure for sample preparation was described previously (Satoh et al, 2010). For fractionation, two C57BL/6J mouse hypothalami were dissected into buffer A (10 mM HEPES-KOH, pH 7.9, 10 mM KCl, 1.5 mM $MgCl_2$, 0.5 mM DTT, 1 mM PMSF, protease inhibitor cocktail [Roche], 1 mM NaF, 1 mM $Na_3VO_4$). The tissue was incubated on ice to swell 20 min. Samples were homogenized 10 s on medium speed using a Polytron homogenizer and centrifuged 3 min at 1,500$g$. The resulting cytoplasmic supernatant (S1) was removed. The remaining pellet was washed with buffer A and centrifuged again. The washed pellet was resuspended in buffer B (20 mM HEPES-KOH, pH 7.9, 400 mM NaCl, 1.5 mM $MgCl_2$, 200 mM EDTA, 0.5 mM DTT, 1 mM PMSF, protease inhibitor cocktail, 1 mM NaF, 1 mM $Na_3VO_4$) with 200 $\mu$g/ml RNaseA and the sample was rocked 30 min at room temp and centrifuged 3 min at 18,000$g$. The resulting RNase-extractable supernatant (S2) was removed. The remaining pellet was washed with buffer B and centrifuged again. The washed pellet was resuspended in buffer B with 300 $\mu$g/ml DNase I (QIAGEN) and 5 mM $MgCl_2$. The sample was incubated 30 min at 37°C, mixing periodically, and then centrifuged 3 min at 18,000$g$. The resulting DNase-extractable supernatant (S3) was removed. The remaining pellet was washed with buffer B and centrifuged again. 2× Laemmli's sample buffer was added to the insoluble pellet (P), homogenized with syringe and 28 G needle, and boiled 5 min. After centrifugation, no pellet remained. To make protein extracts from S1 to S3 fractions, 5× Laemmli's sample buffer was added and samples were boiled 5 min. The 8% equivalent of each fraction by volume was run on a 4–15% TGX gel (Bio-Rad) for Western blotting using affinity-purified polyclonal rabbit anti-mouse Prdm13. Antibodies for Western blotting included affinity-purified polyclonal rabbit anti-mouse htPrdm13 (Covance), anti-GAPDH antibody (MAB374MI; Thermo Fisher Scientific) and anti-histone H3 antibody (#9715; Cell Signaling Technology).

## Western blot analysis of Prdm13 using Prdm13-PA-Tag hypothalami

Hypothalami from *Prdm13*-PA-Tag KI C57BL/6J mice or WT C57BL6J mice were dissected and frozen in liquid nitrogen. One mouse brain was homogenized using a syringe and needle in 60 $\mu$l of lysis buffer A (10 mM HEPES, pH 7.5, 2 mM $MgCl_2$, 3 mM $CaCl_2$, 300 mM sucrose, 1 mM DTT) supplemented with Halt protease and phosphatase inhibitor cocktail (Thermo Fisher Scientific). The homogenates were incubated 10 min on ice and centrifuged at 600 $g$ for 5 min. The supernatants were transferred to new tubes as cytoplasmic fractions. The pellets were resuspended in 60 $\mu$l of lysis buffer B (50 mM HEPES, pH 7.4, 150 mM NaCl, 2.5% SDS, 2 mM $MgCl_2$, 1 mM DTT) supplemented with Halt protease and phosphatase inhibitor cocktail and homogenized using a syringe and needle. The homogenates were centrifuged at 16,000 $g$ for 20 min. The supernatants were transferred to new tubes as nuclear fractions. Protein concentration was determined using the BCA protein assay kit (Takara Bio).

Equal amounts of protein extracts were resolved by SDS–PAGE using 4–15% Mini-PROTEAN TGX precast gel (Bio-Rad) and transferred to PVDF membrane. Prdm13-PA was detected using anti-PA-Tag antibody conjugated with HRP (015-25951; Fujifirm). Histone H3

and GAPDH were detected using anti-histone H3 antibody (#9715; Cell Signaling Technology) or anti-GAPDH antibody (MA5-15738; Thermo Fisher Scientific) as primary antibodies, and anti-rabbit IgG HRP–linked whole antibody (NA934V; GE Healthcare) or anti-mouse IgG HRP–linked whole antibody (NA931V; GE Healthcare) as secondary antibodies. Protein bands were visualized using Amersham ECL Select (Cytiva).

## Plasmid construction for reporter assay

Hypothalamic *Prdm13*–coding sequence was amplified from a cDNA of C57BL/6J mouse hypothalamus using primers containing a FLAG-tag–coding sequence. The amplified DNA fragment was introduced into pcDNA3.1(+) mammalian expression vector (Thermo Fisher Scientific) using *Eco*RI and *Xba*I sites, creating pcDNA3.1-Prdm13 plasmid, which expresses C-terminal FLAG-tagged hypothalamic *Prdm13* driven by the CMV promoter. A plasmid expressing FLAG-tagged Prdm13-202 or mutants of Prdm13-202 was constructed as follows. The coding sequence of amino acids (AA) 99-754 of Prdm13-202 and C-terminal FLAG-tag was PCR amplified from the pcDNA3.1-Prdm13 plasmid. The amplified DNA fragment and a DNA fragment coding AA 1-98 of Prdm13-202 synthesized as a gBlocks Gene Fragment (Integrated DNA Technologies) were assembled into FCIV.FM1 vector using NEBuilder HiFi DNA Assembly Master Mix (New England Biolabs), creating a plasmid expressing C-terminal FLAG-tagged Prdm13-202 driven by a ubiquitin promoter. Similarly, the coding sequence of AA 181-754 of Prdm13-202 was amplified and introduced into FCIV.FM1, creating a plasmid expressing the PR-domain deletion mutant of Prdm13-202, named Prdm13-deltaPR. Prdm13-202 with amino acid mutations (C187A, H207A, C622A, H638A, C650A, H666A, C679A, H695A) leading to inactivation of four zinc finger domains (Chang et al, 2013) named Prdm13 mutZif. A plasmid expressing Prdm13 mutZif was created by introducing DNA fragment coding Prdm13 mutZif synthesized as a gBlocks Gene Fragment (Integrated DNA Technologies) into FCIV.FM1 vector. Luciferase reporter vectors were constructed as follows. Approximately 4 kbp upstream sequence from the transcription start site of mouse *Cck*, *Grp*, or *Pmch* gene was PCR amplified and inserted into pGL4.1 (Promega) using *Kpn*I and *Hin*dIII sites.

## Gene expression analysis of DMH samples

The compact region of the DMH was collected by laser microdissection using the Leica LMD 6000 system (Leica). The detailed procedure for sample preparation was described previously (Satoh et al, 2010). Total RNA was extracted following laser microdissection using the PicoPure RNA isolation kit (Applied Biosystems). cDNA was synthesized using the Applied Biosystems High-Capacity cDNA Reverse Transcription Kit. Quantitative real-time RT-PCR was conducted, and relative expression levels were calculated for each gene by normalizing to *Gapdh* levels and then to the average of the control samples. Primers used in this study were Mm99999915_g1 (*Gapdh*), Mm00446170_m1 (*Cck*), Mm00612977_m1 (Grp), Mm01242886_g1 (*Pmch*), and Mm01217509_m1 (*Prdm13*) (Applied Biosystems).

## Reporter assay for *Prdm13* transcriptional variants and mutants

250 ng of the reporter plasmid and 250 ng of the expression plasmid were transiently co-transfected into NIH3T3 cells (Nakamura et al, 2011) (a gift from Dr. Sugimoto) using HilyMax transfection reagent (Dojindo). For mock assay, 250 ng of empty FCIV.FM1 vector was used instead of the expression vector. After 24 h, luminescence was measured using the dual luciferase reporter system (Promega) without detecting renilla luciferase activity. Obtained luminescence was normalized to total protein concentration measured by BCA protein assay kit (Takara Bio).

## Wheel-running analysis

Mice were individually housed into cage with the wireless running wheel (Med Associates Inc.) and habituated for 2 wk. Basal physical activity was recorded with Wheel Manager software (Med Associates Inc.) for 4–5 d under 12/12-h light/dark cycle. After the basal measurement, physical activity recorded under constant darkness for 10 d. Physical activity and period length were determined by Wheel Analysis software (Med Associates Inc.). Light/dark cycles were strictly monitored by a light censor (Brain Science Idea Co. Ltd.).

## Measurement of adipocyte size

Mice were anesthetized with isoflurane and perfused with PBS followed by 4% PFA. The perigonadal WAT were fixed with 4% PFA overnight and placed into 70% ethanol. Paraffin sections were prepared by Tissue Tech VIRTM 5 Junior (VIP-5-Jr-10; Sakura Fine Chemical), and HE-staining was conducted by Multiple Slide Stainer (DRS 2000-B; Sakura Fine Chemical). Slide images were scanned by NanoZoomer (Hamamatsu Photonics). The sections were viewed at 20x magnification and randomly selected five areas each section using Nano-Zoomer NDP.view2 (Hamamatsu Photonics). The size of adipocytes was measured by ImageJ with Adipocyte_Tools.ijm. Genotypes and conditions were blinded during the experimental procedures.

## Longevity study

All animals were kept in our animal facility with free access to standard laboratory diet and water. No mice used for the longevity study were used for any other biochemical, physiological, or metabolic tests. The endpoint of life was the time when each mouse was found dead during daily inspection. Moribund mice were euthanized according to our institutional animal care guidelines, and the time at euthanasia was its endpoint. Survival data of each cohort were analyzed by plotting the Kaplan–Meier curve and performing the log-rank test using Prism. Tumor and organ tissues were dissected immediately after the animals were euthanized or dead, fixed with 10% formalin neutral buffer solution (Fujifilm) for 24 h, and processed for paraffin-embedded sections, followed by hematoxylin–eosin staining for pathological diagnosis. Immunostainings were performed, if necessary, for differentiation of tumors, by BOND MAX/III (Leica) with Bond Polymer Refine Detection (ds9800; Leica). Antibodies against CD68 (1:2,000 dilution; histiocytic marker; E3O7V; #97778; Cell Signaling Technology), alfa-fetoprotein (1:500 dilution; 14550-1-AP; Proteintech), CD45R (1:300 dilution; B-cell marker; B220; #550286; BD Biosciences; with F [ab'2] anti-rat IgG [H&L]; 1:500 dilution; 712-4126; Rockland antibodies & assays), and CD3 (1:500 dilution; T-cell marker; 21120-1-AP; Proteintech) were used.

## Statistical analysis

Excel or GraphPad Prism 9 software was used for data quantification and generation of graph. One-way ANOVA followed by Bonferroni's post hoc test was used for comparisons between three or more groups. Repeated measures ANOVA followed by Bonferroni's post hoc test was used for number of bouts or episode duration every 3 h for a 24-h period, EEG SWA for a 24-h period and after SD, number of sleep attempts during SD, amount of wakefulness, NREM sleep, and REM sleep for a 24-h period and physical activity. Two-way ANOVA followed by Bonferroni's post hoc test was used for testing the differences between the age groups and experimental conditions. FFT significance was determined by two-way ANOVA. Number of bouts and episode duration significance at each period (light or dark period), total amount of wakefulness, NREM, or REM sleep each period (light, dark, or total 24-h period) was determined by unpaired $t$ test. Significance of the number of sleep attempts for 2 h before/after injection taken from the same individual was determined by paired $t$ test. Pierson's correlation was used for examining the relationship between number of bouts and number of sleep attempts, or between total number of sleep attempts and remaining lifespan. Log-rank test was used for longevity study.

# Supplementary Information

# Acknowledgements

A Satoh is supported by grants from JSPS KAKENHI (17H07417, JP18H03186, 20K21780, 22H04987), the Japan Agency for Medical Research and Development (AMED) (JP20gm5010001s0604), the America Academy of Sleep Medicine Foundation, the Takeda Science Foundation, and the Research Fund for Longevity Sciences from the NCGG (28–47). S Tsuji is supported by grants from JSPS KAKENHI (22K17848) and Suntory Foundation for Life Science. M Wong and N Rensing are supported by a grant NIH P50 HD103525 to the Washington University Intellectual and Developmental Disabilities Research Center. S Imai is supported by grants from the National Institute on Aging (AG037457, AG047902) and the Tanaka Fund at Washington University School of Medicine and also by AMED (JP20gm5010002s0003) at the IBRI. K Nakamura is supported by grants from AMED (JP20gm5010002s0304) and JST Moonshot R&D (JPMJMS2023). T Furukawa is supported by grants from AMED (JP21gm1510006), JST Moonshot R&D (JPMJMS2024), and JST (JPMJPF2018). S Toyokuni is supported by a grant from AMED (JP21gm5010003). S Mizuno and S Takahashi are supported by a grant from AMED (JP18gm5010003). This work was performed in part under the Collaborative Research Program of Institute for Protein Research, Osaka University.

## Author Contributions

S Tsuji and A Satoh: conceptualization, investigation, methodology, and writing—original draft.

CS Brace: conceptualization, investigation, methodology, and writing—review and editing.

R Yao: conceptualization, investigation, and methodology.

Y Tanie: conceptualization and methodology.

H Tada: investigation and methodology.

N Rensing, K Nakamura and M Wong: methodology and writing—review and editing.

S Mizuno, J Almunia, Y Kong, T Furukawa, N Ogiso, S Toyokuni, and S Takahashi: methodology.

S Imai: conceptualization, methodology, and writing—review and editing.

## Conflict of Interest Statement

S Imai receives a part of patent-licensing fees from MetroBiotech (USA), Teijin Limited (Japan), and the Institute for Research on Productive Aging (Japan) through Washington University. All other authors declare no competing interest.

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
