## [Reviewer comments · Life Science Alliance]

Manuscript number: RC-2022-01686

Corresponding author(s): Akiko, Satoh

1. General Statements [optional]

This section is optional. Insert here any general statements you wish to make about the goal of the study or about the reviews.

In this study, we report the potential role of Prdm13 in the DMH on age-associated sleep alterations and DR effects against aging. Our study demonstrated that DMH-Prdm13-KO mice recapitulate some age-associated sleep alterations, such as sleep fragmentation and excessive sleepiness during SD. DR nicely rescued these age-associated sleep alterations, and the rescue requires the existence of Prdm13 in the DMH, revealing the importance of Prdm13 in the DMH for age-associated physiological changes. Additionally, our new results clearly showed that chemogenetic inhibition of Prdm13+ DMH neurons is able to recapitulate excessive sleepiness during SD, which is one of key age-associated sleep alterations. This new finding significantly strengthens our conclusion that Prdm13 in the DMH neurons plays an important role in the regulation of age-associated sleep alterations. We also found that Prdm13 works as a transcription factor in the hypothalamus and its targets are potentially involved in the regulation of age-associated sleep alterations.

Reviewer #1 (Evidence, reproducibility and clarity (Required)):

In this manuscript, the authors report dorsomedial hypothalamus-specific PR-domain containing protein 13-knockout (DMH-Prdm13-KO) mice recapitulated age-associated sleep alterations such as sleep fragmentation and increased sleep attempts during sleep deprivation (SD). These phenotypes were further exacerbated during aging, with increased adiposity and decreased physical activity, resulting in shortened lifespan. Moreover, overexpression of Prdm13 in the DMH ameliorated sleep fragmentation and excessive sleepiness during SD in old mice. They identified maintaining Prdm13 signaling in the DMH might play an important role to control sleep-wake patterns during aging. These findings are interesting and novel and the evidence they provided looks solid.

We deeply appreciate that this reviewer found our findings are interesting and the evidence solid.

Major comments

1. The author spent a lot of words on Sirt1 in the introduction. Since Sirt1 regulates Prdm13, is there a link between the two in age-related sleep changes? If so, you can add some results and discussion.

Thank you very much for raising this important issue. Our previous study demonstrated that a mouse model with high hypothalamic Sirt1 activity displays reduced number of transitions between wakefulness and NREM sleep (reference # 15), revealing that hypothalamic Sirt1, as well as Prdm13, is involved in the regulation of sleep fragmentation. However, sleep propensity was not altered in *Sirt1*-overexpressing transgenic mice (reference #13) and DMH-Prdm13-KO mice (Fig. 1). Based on these findings, we added the following sentence in the Results.

On page 12, line 308-318

"..... Similarly, a mouse model with high hypothalamic Sirt1 activity displays reduced number of transitions between wakefulness and NREM sleep¹⁵, revealing that hypothalamic Sirt1, as well as Prdm13, is involved in the regulation of sleep fragmentation. Sleep propensity was not altered in Sirt1-overexpressing transgenic mice¹³. Given that the level of hypothalamic Prdm13 and its function decline with age, age-associated sleep fragmentation could be promoted through the reduction of Prdm13/Sirt1 signaling in the DMH, but sleep propensity might be increased by other mechanisms."

2. In Figure 2e, the author describes n=7-8 in the figure legend, but why do both groups on the column show eight data? Is there something wrong with the statistics? Please check the statistics in the article carefully.

We corrected n=7-8 to n=8 in the figure legend of Fig. 2e.

3. DMH is known as one of the major outputs of hypothalamus circadian system and is involved in the circadian regulation of sleep-wakefulness (*J.Neurosci.* 23, 10691-10702 ; *Nat Neurosci* 4:732-738). Does Prdm13 correlate with circadian rhythms? The author can add relevant content to the discussion

As per this reviewer's suggestion, we added the following sentence in the Discussion on page 22, line 614-618,

"For instance, it would be of great interest to elucidate whether Prdm13 signaling in the DMH contributes to regulate the circadian system, since the DMH is known to be involved in the regulation of several circadian behaviors^{32,33}. Although DMH-Prdm13-KO mice did not display abnormal period length compared with controls, further studies are needed to address this possibility."

Minor comments

1. The immunohistochemical diagram in the paper is not representative enough, as shown in FIG. 2b and c.

We apologize that our presentation in Figs. 2a-c was confusing. Although Fig. 2b shows the numbers of cFos cells in the entire region of the DMH (summed up from three DMH regions), the images in Fig. 2c are from one of DMH regions for each condition. To avoid confusion, we revised the legend of Figs. 2a-c and the manuscript in the Results as follows:

-In the figure legend of Figs. 2a-c

"a, Total numbers of cFos+ cells b,c, Images of DMH sections at bregma -1.67 mm"

-In the Results on page 7, line 189

"..... the hypothalamus, the DMH (summed up from bregma -1.67 to -1.91mm) showed a greater number of cFos+ cells during SD compared to SD-Cont (Fig. 2a-c, Supplementary Fig. 2a)....."

2. In FIG. 5h, the authors showed that the effect of overexpression of Prdm13 was very obvious, but the expression range of the virus after injection was lacking. Is there a fluorescent gene such as GFP on the virus to directly see the expression of the virus in the brain?

Unfortunately, we do not hold extra samples to check the distribution of the virus after injection. However, we have established sufficient injection technique to target the DMH using the lentivirus system that we used in this study (Satoh et al *Cell Metab* 2013).

3. Were mice singly housed or housed in groups?

Most of the mice were housed in groups, except for the DR study. We added this information in the section Animal models of the Methods on page 44, line 1072

"....RIKEN BRC. Most of the mice were housed in groups, except for the DR study. For the DR study ,..... "

4. *The part of sleep analysis needs to be further refined. How can REM and NREM in mice be distinguished and according to what criteria?*

We added the criteria to define NREM and REM in the section Sleep analysis of the Methods on page 45, line 1132-1135.

".....with visual examination. EEG periods dominated by higher amplitude delta wave activity with nuchal muscle atonia were scored as NREM sleep epochs. REM sleep consisted of periods of semi-uniform theta activity EEG with muscle atonia and/or muscle atonia with brief myoclonic twitches. Score was blinded"

5. *The authors may consider adding more recent literature related to DMH and sleep, such as DOI: 10.1093/cercor/bhac258*

We incorporated this reference to the following sentence in the section Results on page 8, line 203.

"..... Although DMH neurons are linked to sleep²¹, aging and longevity "

Reviewer #2 (Evidence, reproducibility and clarity (Required)):

Summary:

In this study, Tsuji et al. demonstrate that Prdm13 signaling is involved in the regulation of sleep-wake pattern. They also identified Prdm13 as a transcription factor in the DMH neurons.

Major comments:

1. *The evidence presented in Fig. 1 of age-related sleep fragmentation is potentially problematic. Although many previous studies have demonstrated fragmented sleep, especially fragmentation of NREM sleep, in aged mice compared to young mice, the data here do not suggest NREM fragmentation, because no change in the NREM bout duration was found. REM, on the other hand, may indeed have fragmentation during the dark phase, but REM only takes a small portion of the total sleep. Therefore, the conclusion that sleep is fragmented in old mice is not fully supported by Fig. 1. I noticed that the authors used 4-6 months old mice as the young group. Mice of this age can hardly be called "young". The females even start to have lowered fertility. This might be one of the reasons for the discrepancy between this and other studies. Repeating these experiments (and others involving the young group) with mice of more appropriate age (usually 2-3 months old) is recommended. Nonetheless, aging-caused sleep change is not new knowledge and has been reported repeatedly. This part of the results should be in the supplementary figures.*

We deeply appreciate this reviewer's comment. In accordance with this reviewer's suggestion, we carefully reconsidered the age of young mice. Most of published studies used mice at 2 to 4 months of age as the young group [2 to 4-month-old (7 studies), 4.6-month-old (1 study), 6-month-old (1 study), 2 to 6-month-old (1 study)]. Thus, to strictly use mice at 3-4 months of age as the young group, we excluded data of one cohort using mice at 6 months of age (2 mice each age group). Consistent with many previous studies, our revised data demonstrated that sleep fragmentation during NREM sleep is predominantly observed in old mice compared with young mice, particularly during the dark period. Based on these new results, we revised Fig.1, Suppl Fig.1, and all description related to Fig. 1 (manuscript on page 5-7, line 109-180). We would like to keep Fig. 1 as it is. Since most of the previous studies used males but not females, data from

females are still lacking in the field (Campos-Beltran and Marshall, Pflugers. Arch., 473:841-851, 2021).

2. The sleep phenotypes in aged mice and in Prdm13-KO mice are clearly distinct from each other. In the old mice (Fig. 1), REM sleep is fragmented but the total amount remains unchanged, and NREM sleep is increased (both bout number and total amount), indicating there may be more REM-to-NREM transitions, which the authors should quantify. However, Fig. 3 shows in Prdm13-KO mice, there is no REM fragmentation. In fact, it even seems to stabilize REM. But NREM duration is shortened, and no change in the total NREM or REM sleep time. These results suggest that the sleep alterations caused by aging and Prdm13-KO might have some overlap but are mostly in parallel and likely through different mechanisms. Therefore, the rationale of connecting Prdm13 signaling to aging-caused sleep changes is questionable. Is there a developmental change of Prdm13 expression in DMH between young and old mice? The authors also showed that Prdm13-KO in old mice caused decrease in NREM duration but has no effect on REM sleep, but in normal old mice, it is REM, but not NREM that has a defect. Prdm13 overexpression also only mildly decreased NREM bout number without affecting the episode duration of either NREM or REM, which can hardly be interpreted as "ameliorating sleep fragmentation". To me, all these results just suggest parallel actions of Prdm13 and aging on sleep, with Prdm13 mostly affecting NREM sleep but aging mostly impairing REM sleep.

We deeply appreciate this reviewer's keen eyes. We carefully reassessed REM sleep data in new Fig. 4. The revised data showed that whereas the duration of NREM episodes in DMH-Prdm13-KO mice during the dark period were significantly shorter compared to control group, the duration of REM episodes in the KO mice was not significantly altered. Therefore, after revising Fig. 1 and 4, our results showed that both aging and Prdm13-KO similarly affect the duration of NREM sleep episodes. These results suggest that sleep fragmentation, in particular, during NREM sleep, is commonly observed in old mice and DMH-Prdm13-KO mice. In addition to sleep fragmentation during NREM sleep, excessive sleepiness during SD was also commonly observed in old mice and DMH-Prdm13-KO mice. Additionally, we have recently conducted a chemogenetic (DREADD) experiment to test whether inhibition of Prdm13+ DMH neurons recapitulates phenotypes of aged mice during SD, since Prdm13+ DMH neurons get activated during SD, compared with a normal sleep state, in young mice but not old mice. Chemogenetic inhibition of Prdm13+ DMH neurons was able to recapitulate excessive sleepiness during SD, which is one of key age-associated sleep alterations. This new finding significantly strengthens our conclusion that Prdm13 in the DMH neurons plays an important role in the regulation of age-associated sleep alterations (new data shown in Fig. 3). On the other hand, the effect of aging and Prdm13-KO on sleep propensity was distinct from each other. We think that age-associated sleep fragmentation could be promoted through Prdm13 signaling in the DMH, but sleep propensity might be increased by other mechanisms. We described these results and possibilities in the Results, and revised the Abstract as follows:

On page 12, line 308-318

"activity in DMH-Prdm13-KO mice (Fig. 4h, Supplementary Fig. 4f-h). Together, sleep fragmentation during NREM sleep and excessive sleepiness during SD are commonly observed in old mice and DMH-Prdm13-KO mice, but the effects of aging and Prdm13-KO on sleep propensity ... Given that the level of hypothalamic Prdm13 and its function decline with age¹⁶, age-associated sleep fragmentation could be promoted through the reduction of Prdm13/Sirt1 signaling in the DMH, but sleep propensity might be increased by other mechanisms."

On page 2, line 49

"Dietary restriction (DR), a well-known anti-aging intervention in diverse organisms, ameliorated age-associated sleep fragmentation and increased sleep attempts during SD, whereas these effects of DR were abrogated in DMH-*Prdm13*-KO mice."

We also added new Figures (Fig. 2h, Supplementary Fig. 2h, Fig. 3a-e, Supplementary Fig. 3) and description related to new data from chemogenetic study in the Abstract, Introduction, Results, Discussion, and Materials and Methods.

On page 2, line 39-44,

"... Here, we demonstrated that PR-domain containing protein 13 (*Prdm13*) + neurons in the dorsomedial hypothalamus (DMH) are activated during sleep deprivation (SD) in young mice but not old mice. Chemogenetic inhibition of *Prdm13*+ neurons in the dorsomedial hypothalamus (DMH) in young mice promotes increase in sleep attempts during sleep deprivation (SD), suggesting its involvement in sleep control. Furthermore, DMH-specific *Prdm13*-knockout"

On page 4, line 100-102,

"... Chemogenetic inhibition of *Prdm13*+ neurons in the DMH in young mice promotes increase in sleep attempts during SD, suggesting its involvement in sleep control.... "

On page 9-10, line 228-254,

".....sleep loss during SD in young mice. On the other hand, the percentage of cFos+ cells in *Prdm13*+ DMH neurons during SD did not differ from its percentage during SD-Cont in old mice (Fig. 2h, Supplementary Fig. 2h). These results indicate that normal neuronal activation of *Prdm13*+ DMH neurons to SD is impaired with aging.

Chemogenetic inhibition of *Prdm13*+ DMH neurons induces excessive sleepiness during..
Next, using *Prdm13*-CreERT2 mice and the inhibitory designer receptor exclusively activated by designer drug (DREADD) hM4Di, was comparable between groups. "

On page 24, line 671-674,

".....Chemogenetic manipulation of *Prdm13*+ neurons in the DMH reveals that the inhibition of this neuronal population promotes excessive sleepiness during SD, demonstrating the direct involvement of *Prdm13*+ DMH neurons in controlling sleep-wake patterns through neuronal activity....."

As this reviewer pointed out, the effect of *Prdm13* overexpression on NREM sleep fragmentation seems to be moderate, but we still observed effects on excessive sleepiness during SD. Thus, we revised the manuscript related to *Prdm13*-overexpression study in the Abstract and Results as follows:

On page 2, line 51

"Moreover, overexpression of *Prdm13* in the DMH ameliorated ~~sleep fragmentation and~~ increased sleep attempts during SD in old mice."

On page 17-18, line 466-491

Overexpression of *Prdm13* in the DMH partially affects age-associated sleep alterations
..... (Fig. 6h). The number of wakefulness and NREM sleep episodes in old *Prdm13*-OE mice were significantly lower, whereas duration of wakefulness in old *Prdm13*-OE mice tended to be longer than old control mice during the dark period with no change in the duration of NREM

episodes (Fig. 6i,j). Intriguingly, Thus, the restoration of Prdm13 signaling in the DMH partially rescue age-associated sleep alterations, but its effect on sleep fragmentation is moderate."

3. What is the control manipulation for sleep deprivation? The authors need to clarify this in the Methods. Also, sleep deprivation has confounding effects including but not limited to stress, food deprivation (since food was removed during SD), human experimenter (since a gentle-touch method was used). Without proper controls for these variables, the authors should avoid concluding that the changes they saw at cellular level are due to sleep loss.

Thank you very much for this suggestion. We added detailed description for AL-SD (the control manipulation for SD) in the section SD study of the Materials as follows:

On page 45-46, line 1151-1157

"Mice for control manipulation (AL-SD) were also individually housed prior to the experiment without SD and food removal. We checked the level of blood glucose in the SD study, and found that the level of blood glucose was indistinguishable between SD and AL-SD groups (126 ± 6 and 131 ± 4 mg/dL, respectively), revealing that nutritional status is equal between these two groups."

4. Identification of Prdm13+ cells using neuronal markers should be performed in addition to electrophysiological characterizations.

We performed immunofluorescence using anti-MAP2 antibody and confirmed that most Prdm13+ cells are neurons. We added this new result in Suppl Fig. 2g.

5. Figs. 6 and 7 seem very disconnected from the main story. Identification of Prdm13 as a transcription factor is potentially interesting, but how does it account for its role in affecting sleep? The criteria of picking Cck, Grp and Pmch out of other candidate genes potentially regulated by Prdm13 and the rationale to investigate these genes seem unclear. More importantly, no evidence was shown regarding how Cck/Grp

Base on RNA-sequencing using DMH samples from DMH-Prdm13-KO and control mice, we got several candidate genes as downstream genes of Prdm13. After validating the candidate genes by qRT-PCR, Cck, Grp and Pmch were detected as top-hit genes. We thus further assessed these three genes in this study. Our result showed that Cck expression in the hypothalamus significantly declines with age. Based on other literature, hypothalamic Cck seems to be involved in sleep control. Therefore, it is conceivable that Prdm13 controls age-associated sleep alterations via modulating Cck expression. However, as this reviewer pointed out, we are still lacking the evidence showing the role of Prdm13/Cck axis in age-associated sleep alterations. The detailed molecular mechanisms by which Prdm13 in the DMH regulates age-associated sleep fragmentation and excessive sleepiness during SD still need to be elucidated in future study.

Minor

comments:

1. Please note on the images of Fig. 2d what the green fluorescence was. It is very confusing as is, given that it's surrounded by quantifications of c-fos in the figure.

The label "Prdm13" was added in Fig. 2d.

2. Please note use a different color for Prdm13 in several figure images (e.g., Fig. 2f, g, 7a,d, and Supplementary 2c). Yellow usually means overlap of red and green.

Since we have four-color images in Fig. 8, we consistently used yellow for Prdm13 throughout the main figures of the paper. At this moment, we would like to keep the current version of images, but we will revise images if the editor of affiliate journal requests this revision.

3. Please note the statistic test results on power spectrum graphs.

We added the statistic test results on power spectrum graphs in Figs. 1d, 5c, and 6d.

4. Inconsistency between the graphs in Fig. 3d and the description in the text. Fig. 3d suggests no change in Wake episode duration, significant decrease in Dark phase NREM and significant increase in Dark phase REM, whereas lines 224-227 in the main text state "The duration of wakefulness episodes ... was significantly shorter than control mice during the light period, and the duration of NREM sleep episodes ... was significantly longer ... during the dark period (Fig. 3d)". Which one is correct? Please check.

We apologize for this typo and unclear description. We revised the sentence regarding Fig. 4d as follows:

On page 11, line 277-281

"The duration of wakefulness episodes in DMH-*Prdm13*-KO mice was significantly shorter than control mice during the light period between ZT0 to ZT2. The duration of NREM sleep episodes in DMH-*Prdm13*-KO mice was significantly shorter than control mice during the dark period (**Fig. 4d**). These results indicate that DMH-*Prdm13*-KO mice showed ~~mild~~ sleep fragmentation compared with control mice."

5. Fig. 5f, Y-axis title should be EEG SWA.

We corrected it.